# Evidence for dynastic succession among early Celtic elites in Central Europe

Joscha Gretzinger[1], Felicitas Schmitt[2,9], Angela Mötsch[1,9], Selina Carlhoff[1], Thiseas Christos Lamnidis[1], Yilei Huang[1], Harald Ringbauer[1], Corina Knipper[3], Michael Francken[2], Franziska Mandt[2], Leif Hansen[2], Cäcilia Freund[1], Cosimo Posth[4,5], Hannes Rathmann[4,5], Katerina Harvati[4,5,6], Günther Wieland[2], Lena Granehäll[7], Frank Maixner[7], Albert Zink[7], Wolfram Schier[8], Dirk Krausse[2]✉, Johannes Krause[1]✉ & Stephan Schiffels[1]✉

The early Iron Age (800 to 450 BCE) in France, Germany and Switzerland, known as the 'West-Hallstattkreis', stands out as featuring the earliest evidence for supra-regional organization north of the Alps. Often referred to as 'early Celtic', suggesting tentative connections to later cultural phenomena, its societal and population structure remain enigmatic. Here we present genomic and isotope data from 31 individuals from this context in southern Germany, dating between 616 and 200 BCE. We identify multiple biologically related groups spanning three elite burials as far as 100 km apart, supported by trans-regional individual mobility inferred from isotope data. These include a close biological relationship between two of the richest burial mounds of the Hallstatt culture. Bayesian modelling points to an avuncular relationship between the two individuals, which may suggest a practice of matrilineal dynastic succession in early Celtic elites. We show that their ancestry is shared on a broad geographic scale from Iberia throughout Central-Eastern Europe, undergoing a decline after the late Iron Age (450 BCE to ~50 CE).

The European Iron Age north of the Alps is characterized by the two key archaeological cultures Hallstatt (800 to 450 BCE) and La Tène (after 450 BCE until the beginning of the Roman period around 50 BCE), which have been, to a different degree, described as 'Celtic'[1,2]. Today regarded problematic as an ethnonym, the name 'Celtic' was first mentioned in Greek sources from the late sixth century BC, and it is abundantly used in antique sources for societies associated with the La Tène culture[3,4]. Apart from this historical record and its association with the later Hallstatt and La Tène cultures, there is also a connection to linguistic evidence for a common prehistoric language family across large parts of Europe (the Celtic languages). Indeed, the pan-European patterns and linguistic evidence for cultural connections during this time are complex and encompass a vast region from the Iberian Peninsula and the British Isles throughout Central Europe and as far east as Anatolia (during the third century BCE). While older research assumed an exclusive emergence of this later pan-European phenomenon in

[1]Max Planck Institute for Evolutionary Anthropology, Leipzig, Germany. [2]Landesamt für Denkmalpflege im Regierungspräsidium Stuttgart, Esslingen, Germany. [3]Curt Engelhorn Zentrum Archäometrie gGmbH, Mannheim, Germany. [4]Institute for Archaeological Sciences, Department of Geosciences, Eberhard Karls University of Tübingen, Tübingen, Germany. [5]Senckenberg Centre for Human Evolution and Palaeoenvironment, Eberhard Karls University of Tübingen, Tübingen, Germany. [6]DFG Center for Advanced Studies in the Humanities 'Words, Bones, Genes, Tools: Tracking Linguistic, Cultural and Biological Trajectories of the Human Past', Eberhard Karls University of Tübingen, Tübingen, Germany. [7]Institute for Mummy Studies, EURAC Research, Bolzano, Italy. [8]Institut für Prähistorische Archäologie, Freie Universität Berlin, Berlin, Germany. [9]These authors contributed equally: Felicitas Schmitt, Angela Mötsch. ✉e-mail: dirk.krausse@rps.bwl.de; krause@eva.mpg.de; stephan_schiffels@eva.mpg.de

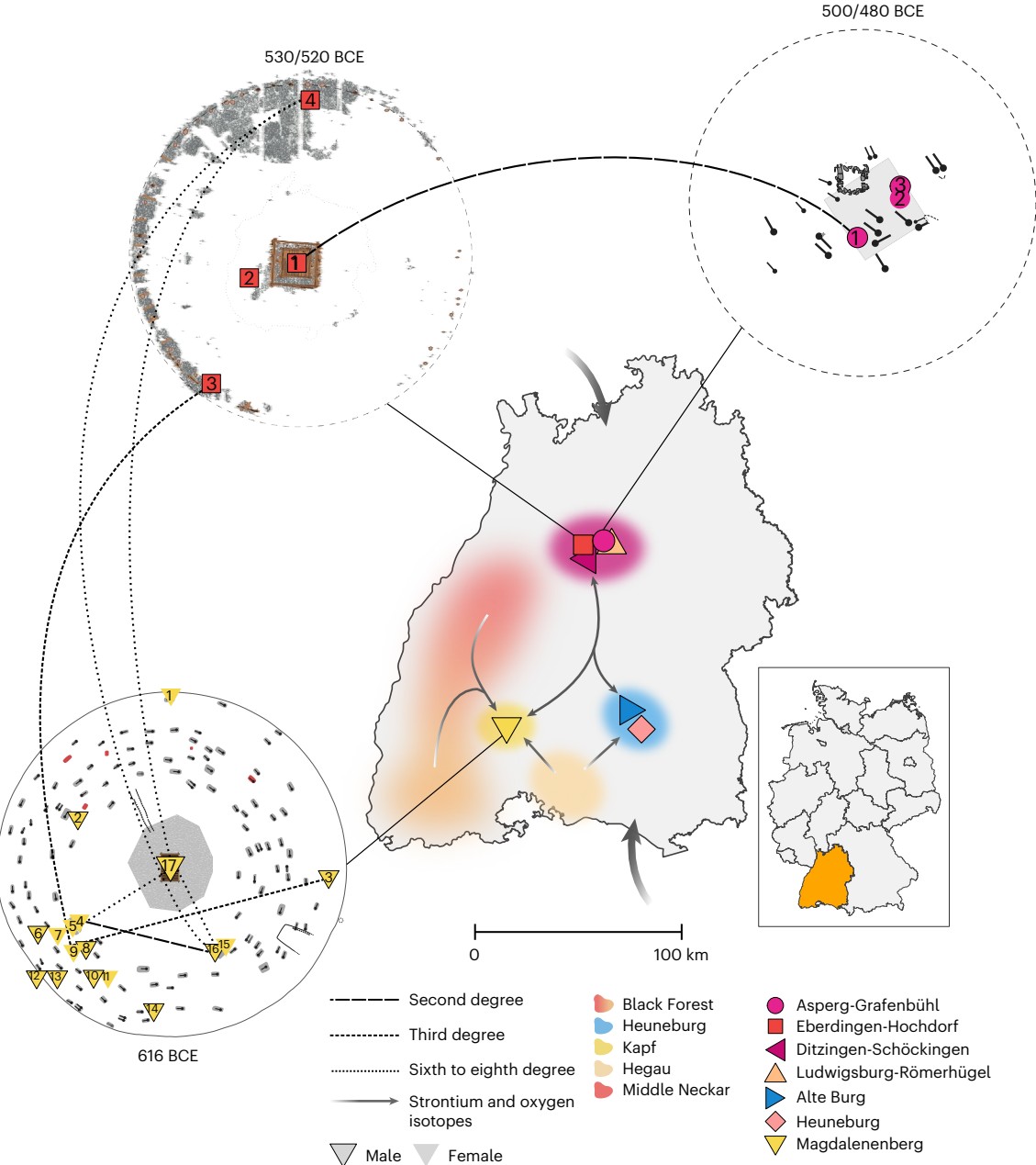

**Fig. 1 | Fine-scale familial relationships and patterns of individual mobility between early Celtic sites.** The map shows the locations of the reported sites in Baden-Württemberg, southwestern Germany (n = 7). The ellipses and arrows on the map indicate the approximate geographical origin areas and general directions of individual mobility based on new and previously published strontium and oxygen isotope values from 67 individuals[37]. Supplementary data can be found in Supplementary Fig. 2.8. Additionally, the site plans of Magdalenenberg (MBG), Eberdingen-Hochdorf (HOC) and Asperg-Grafenbühl (APG) are shown, as well as the dates of their respective central burials (red colour at MBG indicates cremation burials). The sex of the sampled individuals, the respective sample IDs (without site prefixes) and detected familial relationships are indicated. Supplementary data can be found in Supplementary Figs. 2.1–2.3 and Supplementary Tables 2.1–2.4.

a relatively narrowly defined area northwest of the Alps, newer perspectives suggest a model of polycentric emergence in a wide area between the Atlantic coast and southwestern Germany[5]. One of these core regions was located in present-day eastern France, Switzerland and southwestern Germany. Between 600 and 400 BCE (Hallstatt D and La Tène A), this area stands out in its archaeological importance, as highlighted by rich 'princely' burials ('Fürstengräber').

These burials are characterized by monumental burial mounds[6–8] and luxurious grave goods such as ceremonial wagons, furniture, gold jewellery, imported goods from the Greek and Etruscan cultural spheres, or extensive drinking and dining services. Such rare and precious objects have typically been considered indicative of outstanding social status. Throughout the early Iron Age, warrior and sacral-religious representations within those princely burials increasingly conglomerated, merging worldly and spiritual power[9], perhaps more comparable to sacral kings[10,11] rather than mere chieftains[12]. After their death, members of this princely elite were entombed below imposing monuments and became commemorated as heroic ancestors[13,14]. As this development progressed, some of these individuals were buried and worshipped in a god-like manner[11] in large ceremonial complexes, such as the burial monuments near the Glauberg in Hesse, erected in the early La Tène period around 400 BCE[15]. Accordingly,

those monumental princely burials would represent the manifestation of dynastic systems of power, in which political hegemony was at least partially based on biologically inherited privilege[10,11], a hallmark of early complex societies[16].

The nature of the early Celtic political system, especially the importance of biological kinship, has been highly controversial to this day[14]. Some scholars interpret these deceased as 'village elders', who acquired their high social status through personal achievement during their lifetime[17–19] without the precondition of inheritance[20]. The existence of extraordinarily wealthy child burials, indicative of superb social power and prestige, seems to contradict this hypothesis of self-acquired prestige, since those young individuals could hardly achieve such a status during their short lifetime but instead must have inherited it[9]. The argument for hereditary status among elite families is further supported by the recurrent combination of symbols of power such as gold jewellery, precious drinking vessels and wagons associated with the ritual authority of the deceased princes and princesses[12]. A central aspect of a dynastic system of hereditary power is biological relatedness. While there are other forms of kinship, including social relatedness such as fosterage or adoption, which are notoriously difficult to infer from burial archaeology, biological relatedness can be conclusively reconstructed using genetic data. Ancient DNA (aDNA) is therefore a unique tool to address this question but has so far been unsuccessful[21,22]. In this Article, we present genome-wide evidence for the early Celtic society of southwestern Germany and its political organization in the sixth and fifth century BCE.

## Results

### Evidence for dynastic Celtic elites

We selected 31 high-status and secondary burials from seven elite locations in the state of Baden-Württemberg, southwest Germany, namely the large tumulus Magdalenenberg ($n = 17$), the burial mounds of Eberdingen-Hochdorf ($n = 4$), Asperg-Grafenbühl ($n = 3$) and Ludwigsburg-Römerhügel ($n = 3$), the princely burial of Ditzingen-Schöckingen ($n = 1$), the Heuneburg settlement[23] ($n = 2$) and the ritual site Alte Burg[24] ($n = 1$) (Fig. 1 and Supplementary Note 1). For those individuals, we prepared powder from petrous bones and teeth, extracted aDNA and converted it into double-stranded or single-stranded DNA libraries (Methods). We selected all libraries for hybridization DNA capture to enrich aDNA libraries for DNA fragments that overlapped approximately 1.24 million single-nucleotide polymorphisms (SNPs) and generated new genome-wide sequence data for all samples. For the Ditzingen-Schöckingen burial, only the mitochondrial genome was recovered. The final mean coverage at targeted genome-wide SNPs was 0.76-fold (on average, 339k SNPs) with the percentages of endogenous DNA being very low in almost all samples (in the median 0.55%) (Supplementary Table 1.1). We identify 20 individuals as genetically male and 11 as genetically female, with the three central burials of Magdalenenberg (MB017), Eberdingen-Hochdorf (HOC001) and Asperg-Grafenbühl (APG001) being male and the central burial of Ditzingen-Schöckingen being female, supporting the osteological classification (Supplementary Table 1.1). In addition to genome-wide sequences, we measured $\delta^{18}O$ and $^{87}Sr/^{86}Sr$ values for the 17 of those individuals for whom so far no isotope data had been available, to reconstruct patterns of individual mobility (Supplementary Note 2).

Among the individuals studied, we identify several close biological relationships (Fig. 1, Supplementary Note 2 and Supplementary Figs. 2.1–2.3). Most prominently, this includes two of the richest burials in European prehistory, the central graves of Eberdingen-Hochdorf (HOC001) and Asperg-Grafenbühl (APG001), for which we identify a second-degree relationship. Both male individuals share the same mtDNA haplotype J1b1a1 (featuring two private mutations), which suggests relatedness on the maternal line (Supplementary Fig. 2.5). The isotopic data (Supplementary Fig. 2.9 and Supplementary Table 1.2) of the two are very similar, consistent with the biologically available

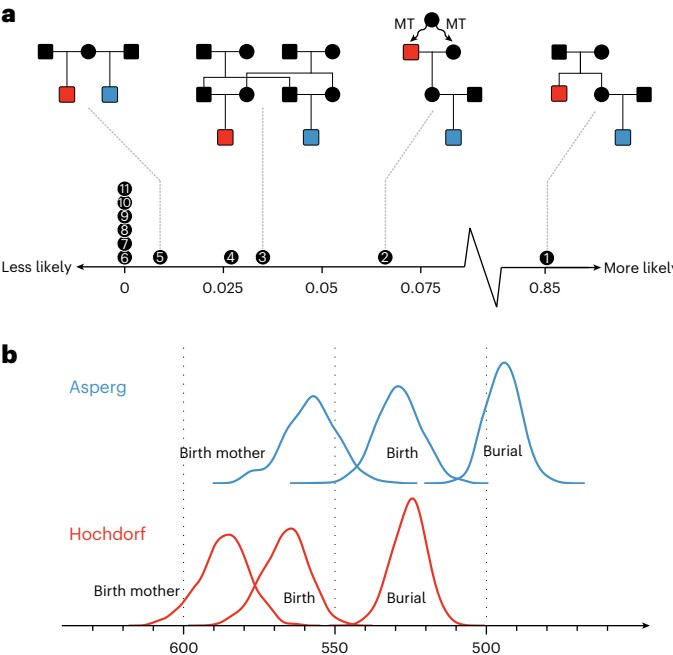

**Fig. 2 | Latent pedigree model connecting the princely graves of Hochdorf (HOC001) and Asperg (APG001). a**, We analyse several plausible pedigrees connecting the two individuals and compute a posterior probability (shown on the *x* axis) given priors from genetic, archaeological and anthropological evidence, including, for example, plausible ages for motherhood (Supplementary Note 3). Females are shown as circles and males as squares; HOC001 is shown in red and APG001 in blue. The labels on the *x* axis correspond to the tested models: (1) HOC001 is the uncle of APG001. (2) HOC001 is the maternal grandfather of APG001, which requires cryptic background relatedness on the mitochondrial lineage. (3) HOC001 and APG001 are double first cousins. (4) HOC001 is the paternal grandfather of APG001. (5) HOC001 and APG001 are half-siblings. (6) HOC001 is the father of APG001. (7) HOC001 and APG001 are full siblings. (8) APG001 is the uncle of HOC001. (9) APG001 is the father of HOC001. (10) APG001 is the maternal grandfather of HOC001. (11) APG001 is the paternal grandfather of HOC001. An avuncular relationship between the two individuals is the most likely scenario, with 86% posterior weight. **b**, Marginal posterior distributions obtained using Markov chain Monte Carlo sampling for burial dates (unobserved but constrained by priors), birth dates as well as the birth date of their respective mother are shown as kernel-density smoothed histograms.

strontium in the middle Neckar region[25] and point to a local origin for both individuals. We integrated archaeological estimates of burial dates, osteological estimates for age at death and multiple lines of genetic evidence (autosomal degree of relatedness, homozygosity and mitochondrial DNA) to derive a Bayesian model for the pedigree that connects both individuals, using latent variables for unobserved family members. Constrained in particular by the distribution of plausible ages of motherhood[26], we obtain marginal posterior probabilities for 11 possible pedigrees consistent with first- and second-degree genetic relatedness and identify an avuncular relationship as the most likely (86%), with HOC001's sister being APG001's mother, compared with a maternal grandparent–grandchild model (6.6%) with HOC001's daughter being APG001's mother, and many less likely scenarios (parents, siblings or cousins) (Fig. 2a and Supplementary Note 3). These results are consistent with previous conjectures about their relationship based on their temporal order and archaeological data[27]. Our Bayesian pedigree model also predicts birth dates and ages of motherhood of unobserved family members (Fig. 2b for the most likely model), allowing a glimpse into the probable life histories of these princely individuals. The close biological relationship between the two may also

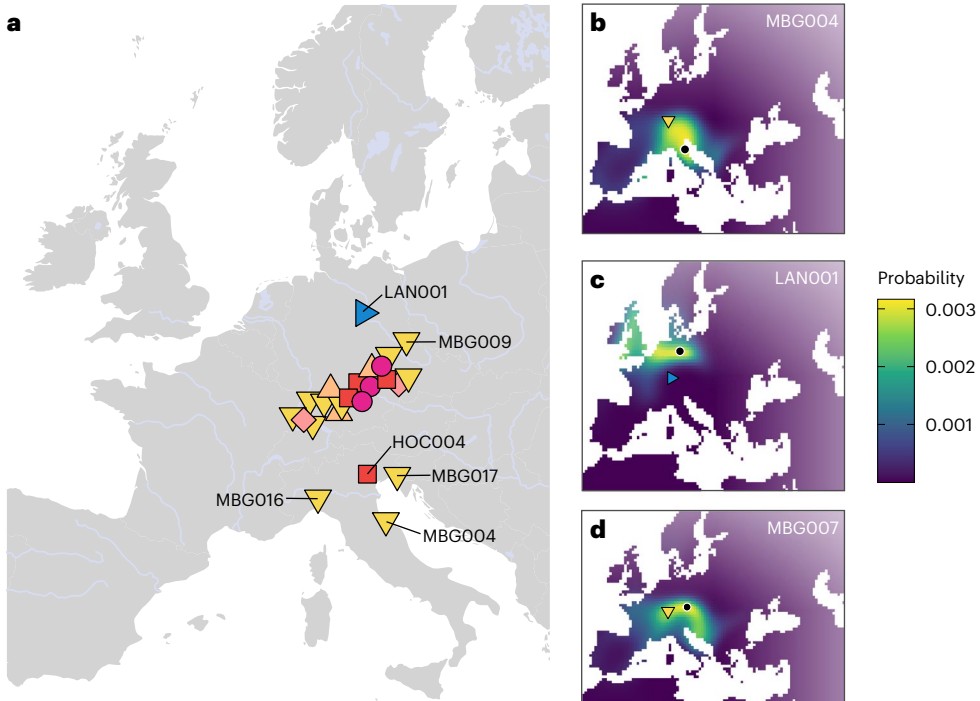

**Fig. 3 | Spatial inferences on the origin of Hallstatt individuals. a**, MOBEST predictions of the geographic regions where the ancestors of Iron Age individuals (*n* = 24) from southwestern Germany originated. Shown are the points of maximum probability at search time 0 (the mean date of the respective individual). The symbols and colours correspond to Fig. 1. **b**, Genetic similarity probability map for MBG004. The filled shape shows the burial location and the black dot the point of maximum probability. **c**, The same as **b** for LAN001. **d**, The same as **b** for MBG007.

explain their exceptional body heights. While male individuals from elite graves are already significantly taller than males from secondary burials (two-sided Wilcoxon rank-sum exact test; *W* = 67, *P* = 0.004067), HOC001, followed by his relative APG001, are the tallest individuals in the complete osteological record of Iron Age southern Germany[10] (Supplementary Fig. 2.10). This highlights the possibility that, besides better nutrition[28,29], also genetic relatedness may have contributed to this social differentiation in body height.

A second unique finding is the long-distance third-degree biological kinship between the richly furnished female MBG009 from Magdalenenberg and the secondary burial HOC003 from Eberdingen-Hochdorf, a pair of relatives spanning more than 100 km and around 100 years (Fig. 1). The mature male HOC003 is not related to any of the other secondary burials or the central grave of the Eberdingen-Hochdorf mound. Consistently, HOC003 shows isotopic values consistent with him being raised in the region around the Kapf, the settlement associated with Magdalenenberg (for details on isotopic results, see Supplementary Note 2 and Supplementary Fig. 2.9), although an origin north of Eberdingen-Hochdorf would also be possible. Such a close inter-site relationship over a large geographic distance is exceedingly rare in the archaeogenetic record (to our knowledge, there is only one comparable case of a second-degree relationship so far[30]). Based on the chronological difference between the graves, an ancestral relationship between both individuals (such as great-grandmother and great-grandson) appears most probable. Within this group of relatives, we additionally identify a third-degree relationship between MBG009 and the young adult male MBG003. Both individuals share the same mtDNA haplotype H1c9, indicating that the close kinship probably derives from the maternal line.

We identified a third inter-site group of relatives, consisting of the two second-degree relatives MBG004 (an adult female) and MBG016 (an adult male), and their more distant relatives MBG017 (the central princely burial) and another secondary burial at Eberdingen-Hochdorf, HOC004, who share identity-by-descent (IBD) fragments typical for relatives of sixth to eighth degree (as inferred using ancIBD[31]; Supplementary Tables 2.6 and 2.7, and Supplementary Fig. 2.4), indicating that all four individuals share a recent common ancestor (Fig. 1). Both MBG016 and MBG004 are exceptional within the burial community: While the sparsely furnished grave of MBG016 is the only grave that overlaps with another burial and is atypically oriented, the grave of MBG004 is extraordinarily wealthy. Both individuals belong to an early phase of the mound and were thus potentially associated with the founding family[32–34]. MBG004 is buried in close vicinity to another female, MBG005, a young adult, who shows no genetic relationship to MBG004 and strontium isotopes typical for the middle Neckar region[25], where the sites of Eberdingen-Hochdorf, Asperg-Grafenbühl and Ditzingen-Schöckingen are located. We note that the biological relatedness detected between the central and secondary burials is consistent with interpretations of the Magdalenenberg as a 'kin group' burial mound for an 'enlarged family'[12].

Interestingly, this third inter-site group of relatives exhibits significantly more southern European ancestry than the rest of our analysed individuals (93.6 ± 1.9% versus 59.9 ± 3.9%; two-sided Wilcoxon rank-sum exact test; *W* = 0, *P* = 0.0002259) and, consequently, significantly more Early European Farmer (EEF) ancestry (55.6 ± 0.9% versus 48.4 ± 1.1%; two-sided Wilcoxon rank-sum exact test; *W* = 0, *P* = 0.0002259) (Supplementary Fig. 2.8) (for details on EEF ancestry decomposition, see Methods and Supplementary Notes 2 and 4). This might indicate a non-local, southern European origin of the ancestors of the Magdalenenberg elite. Consequently, we applied MOBEST[35] to perform spatiotemporal interpolation of their genetic affinity to ~5,660 previously published ancient genomes, obtaining similarity probabilities across early Iron Age Europe that can be interpreted as proxies for geographical origin (Supplementary Note 2). We detect for all four of these samples (MBG004, MBG016, MBG017 and HOC004) a putative transalpine origin in northern Italy, while all other tested Hallstatt individuals' origins are located north of the Alps, close to their respective sites (Fig. 3a,b). Remarkably, these individuals feature excess

EEF ancestry on the X chromosome in comparison with the autosomes (83.5 ± 9.9% versus 55 ± 1.1%). Applying the formula described in Mathieson et al.[36], we find evidence that the EEF admixture was significantly female biased ($Z$ = −2.86), suggesting an excess of females over males with south-European origin among their ancestors. In contrast, we detect no difference in EEF ancestry on the X chromosome and the autosomes in the rest of the sampled Hallstatt population (43.6 ± 5.7% versus 49 ± 0.6%) and, consequently, no evidence for sex-biased admixture in the main group ($Z$ = 0.93).

Zooming into each site, we reconstruct several biological relationships (third to fourth degree) between the secondary burials MBG001 and MBG013, as well as the three burials MBG002, MBG011 and MBG012 (not indicated in Fig. 1; Supplementary Figs. 2.1 and 2.2), which all show isotopic compositions local to the surroundings of the Magdalenenberg and the Black Forest[37] (Supplementary Fig. 2.9). In contrast, none of the secondary burials in Asperg-Grafenbühl and Eberdingen-Hochdorf is related to the respective central graves. Within Asperg-Grafenbühl, we note that the two deceased in the secondary double burial, the adult female APG002 and the male child APG003, are also not biologically related to each other, representing a possible case of fosterage (Discussion). Moreover, APG003 is an outlier in terms of stable isotopes, showing $\delta^{18}$O and $^{87}$Sr/$^{86}$Sr values very similar to the La Tène period male individual LAN001 from a shaft at Alte Burg. While LAN001 most likely originated from coastal northwestern Europe or Central Germany (Supplementary Fig. 2.9), a finding also supported by MOBEST analysis (Fig. 3c), APG003 appears genetically local. His elevated $\delta^{18}$O level may rather reflect breast milk consumption than an origin from a climatically distinct region. Indeed, his $^{87}$Sr/$^{86}$Sr values are very similar to HOC003 and consistent with the biologically available strontium around the Magdalenenberg site, supporting our genetic observation of inter-site mobility. In general, we note that male and female individuals in our sample do not significantly differ in strontium and oxygen isotope values (two-sided Wilcoxon rank-sum exact test; $W$ = 153, $P$ = 0.615 and $W$ = 146, $P$ = 0.4734 for strontium and oxygen, respectively). This stands in contrast to analyses of mobility in Early and Middle Bronze Age southern Germany, where significantly more non-local females than males were found[38]. Furthermore, we do not find a significant association between grave goods, $\delta^{18}$O and aDNA as markers for non-local origin (Supplementary Note 2). For that, we focused on the Magdalenenberg site where a large number of graves exhibit artefacts of transalpine, south-European (especially North Italian and/or southeast Alpine) provenance[37,39], indicating cultural transfer alongside extensive, continuous individual-based mobility. We grouped individuals, for which both isotopes and aDNA data were available, into two groups based on the presence of southern, non-local artefacts. We find that non-local artefacts (being present in 6 out of 16 graves) are not statistically significantly correlated with either higher proportions of EEF ancestry (two-sided Wilcoxon rank-sum exact test; $W$ = 23, $P$ = 0.4923) nor $\delta^{18}$O values (two-sided Wilcoxon rank-sum exact test; $W$ = 44.5, $P$ = 0.1283) (Supplementary Fig. 2.11), both indicating cisalpine origin. Consequently, southern grave goods do not constitute a reliable marker of south-European origin in the Magdalenenberg population, although we do identify individuals with such origins in the burial mound via our isotopic and aDNA data. This is especially evident in the case of MBG010, an adult female, who exhibits $\delta^{18}$O and $^{87}$Sr/$^{86}$Sr values indicative of a northern Italian or Iberian origin[37] yet is neither buried with southern grave goods nor shows excess genetic affinity to those regions (Supplementary Figs. 2.8 and 2.9).

To supplement our findings on biological relationships, we analysed all individuals for evidence of long runs of homozygosity (RoH), which are indicative of consanguinity (a close biological relationship between the parents). We indeed find two individuals with elevated RoH: MBG004 and APG003 (Supplementary Fig. 2.6). Both individuals exhibit over 150c CM of RoH in total, indicative of recent inbreeding, most likely by first cousin parents[40] (Supplementary Fig. 2.7). Given that

such high levels are very rare in the published record, the presence of two consanguineous individuals in the comparably small sample size of 30 individuals may suggest that consanguinity was more frequent among the Hallstatt elites of southwestern Germany than in other ancient societies in the archaeogenetic record.

## Emergence and decline of the West-Hallstatt gene pool

We compared the genome-wide data of our early Iron Age samples with a reference dataset of 5,665 ancient and 10,176 present-day Eurasian individuals (Methods). When projected on the diversity of present-day Europeans by means of principal component analysis (PCA), we find the Iron Age individuals to be separate in genetic space from present-day Germans and falling closer to present-day French and other southern European individuals (Supplementary Fig. 4.1). Compared with contemporaneous data, the Hallstatt individuals cluster homogeneously intermediate between Iron Age samples from present-day France and the Czech Republic[41,42], together with Bronze Age samples from the Bavarian Lech valley[38] within the present-day French variation (Supplementary Figs. 4.2 and 4.3). The divergence between prehistoric and present-day individuals from Germany is also seen in the distribution of genetic distances ($F_{ST}$) (Supplementary Fig. 4.4a) as well as correlation of allele frequencies ($F_4$) (Supplementary Tables 4.6–4.8) on both the population and individual level (Supplementary Figs. 5.1 and 5.4a). The genetic affinity between our Hallstatt individuals from southern Germany and individuals from Bronze and Iron Age France is part of a broader genetic continuum spanning from Iberia to the Balkan peninsula, featuring a common genetic ancestry component (Fig. 4a, green 'CWE' component, Fig. 4b, Supplementary Note 4 and Supplementary Table 4.10).

This broad continuum is characterised by a common demographic process, which we see from an analysis of distal ancestry proportions. In particular, using qpAdm we demonstrate an increase of EEF ancestry and a decrease of Yamnaya and Poltavka pastoralists (OldSteppe) ancestry from the Late Neolithic Bell Beaker period onwards, peaking during the Middle Bronze Age and Iron Age (Supplementary Note 4 and Supplementary Figs. 4.4a and 4.5a) and converging the gene pools in France and southern Germany. This increase of EEF is accompanied by a homogenization of the gene pool in terms of EEF and Steppe ancestry, illustrated by a marked decrease of variance of the per-individual statistic $F_4$(YRI, Test; OldSteppe, EEF) between time periods (Supplementary Fig. 4.4d). This phenomenon was described previously[38,43] and might reflect continuous admixture with coexisting groups in other regions predominantly from southern Europe, who experienced less gene flow from steppe-related populations. It is part of a broader trend of EEF ancestry becoming more similar across central and western Europe in the Bronze Age (Supplementary Fig. 4.5a), coinciding with archaeological evidence of intensified cultural exchange, especially during the Late Bronze Age Urnfield culture period[42]. Indeed, when estimating the time of admixture in individuals from this region ranging from 2500 to 500 BCE (Supplementary Fig. 4.5d), we observe that admixture time decreases significantly with the date of each individual (Spearman's rank correlation, $P$ = 2.98 × 10$^{-7}$), with a slope close to 1.0 (0.75 ± 0.13), which is incompatible with a single pulse of admixture but compatible with stationary continuous and ongoing admixture. To gain insights into the possible sources of this Bronze Age EEF resurgence, we modelled the pooled Hallstatt individuals in qpAdm as a mixture of the Germany_Lech_EBA cluster, and a second source, for which we identify several potential proxies, all of them located in southwestern Europe, especially the Iberian Peninsula and Italy (Supplementary Table 3.12).

To investigate individual ancestries within the Hallstatt group, we used the Middle Bronze Age population from the southern German Lech valley as a proxy for local ancestry. Indeed, most Hallstatt individuals fit a model of receiving all of their ancestry from Germany_Lech_MBA, with the exception of previously described southern outliers

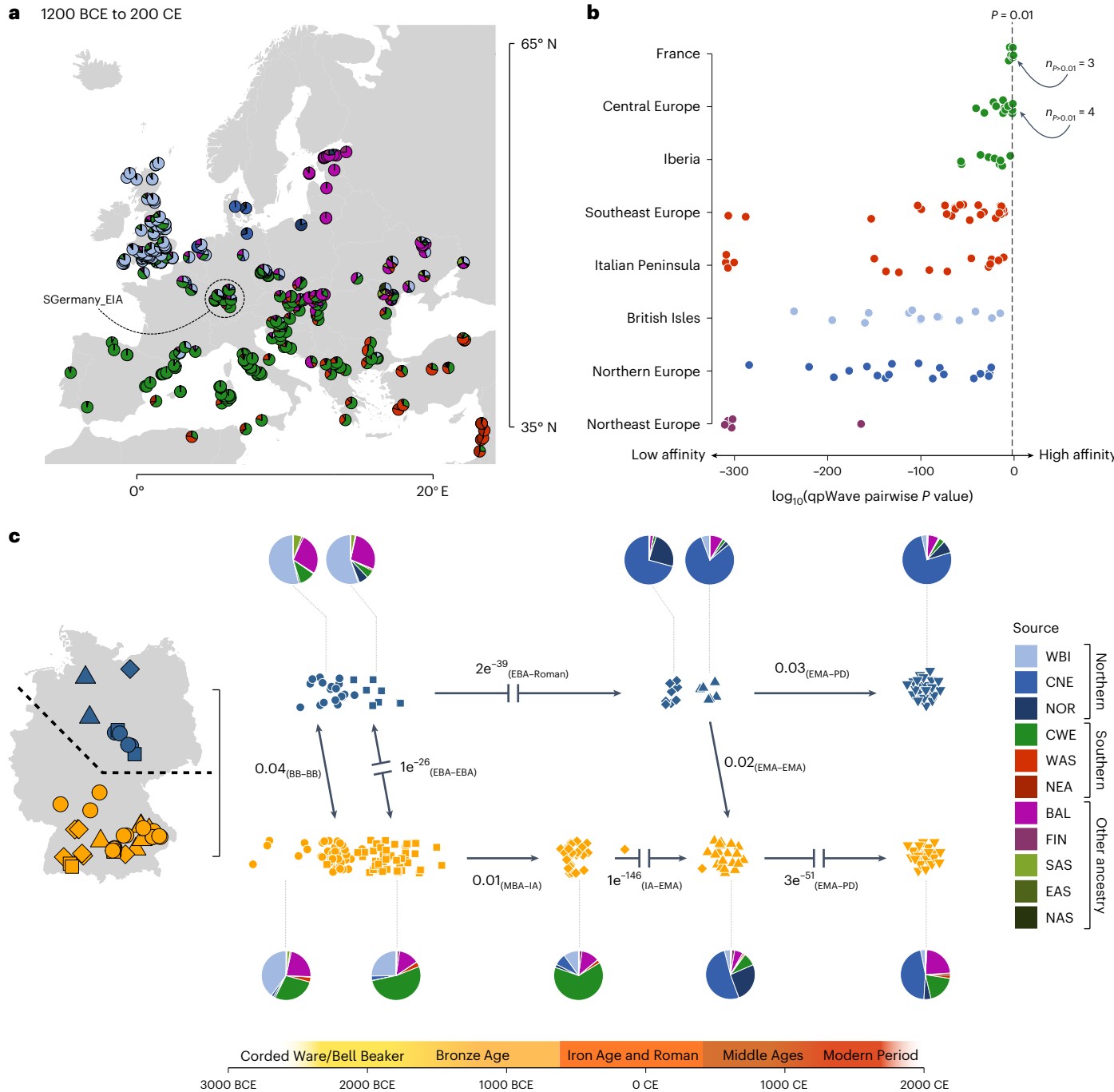

**Fig. 4 | Population genetic affinities across space and time. a**, Mean supervised ADMIXTURE components at $K = 12$ (Supplementary Note 4) aggregated across 5,142 individuals from 342 sites dating between 3,150 and 1,750 years BP. **b**, $P$ values from generalized-likelihood ratio tests implemented in qpWave for testing genetic similarity between southern German Hallstatt individuals and diverse Bronze and Iron Age populations across Europe. Higher $P$ values correspond to higher genetic similarity. **c**, Overview about population genetic changes in Germany from the Late Neolithic to the present day. The arrows indicate $P$ values from generalized-likelihood ratio tests in qpWave for genetic continuity between temporally preceding and succeeding groups in northern Germany (Lower Saxony, Saxony-Anhalt and Mecklenburg-Vorpommern; symbols in blue) and southern Germany (Baden-Württemberg and Bavaria; symbols in orange), respectively (Supplementary Fig. 5.6). Discontinuities are explicitly marked as interrupted arrows. The pie charts depict the averaged ancestry composition derived from supervised ADMIXTURE (Supplementary Note 4) for each group used in qpWave analysis. The sources are WBI (Britain and Ireland), CNE (North Sea zone), NOR (Scandinavia), CWE (Western Europe and Iberia), WAS (Northern Levant), NEA (Southern Levant, Arabia and North Africa), BAL (Baltics), FIN (Finland), SAS (South Asia), EAS (East Asia) and NAS (North Asia).

MBG004, MBG016 and northern outlier LAN001 from Alte Burg (Supplementary Table 2.8). LAN001 received the majority of his ancestry from a more northern European source, most closely related to the Bronze and Iron Age population of the Netherlands and Saxony-Anhalt (Supplementary Tables 2.9 and 2.11), which is also consistent with his elevated $\delta^{18}O$ values supporting a coastal northwestern European or Central German origin[44–46].

The arrival of individuals of more northern European ancestry during the La Tène period can also be observed in published data from the nearby Czech Republic[42], where we analysed individual ancestry

components using supervised clustering (Supplementary Fig. 5.8d) and detect a previously undescribed diversification of the gene pool with respect to northern European ancestry from the Hallstatt to the La Tène period (two-sided $F$ test; $F = 0.20174$, numerator d.f. 15, denominator d.f. 60, $P = 0.001$). In southern Germany (here Baden-Württemberg and Bavaria) the northern European influx broadens to a major genetic turnover between the Iron Age and the Early Middle Ages (Fig. 4c and Supplementary Note 5). It is illustrated by a sharp decrease of EEF ancestry and a substantial resurgence of Steppe-related ancestry together with a re-diversification of the gene pool (Supplementary Figs. 4.4, 4.5 and 5.2). While the Hallstatt population showed highest genetic affinity to present-day French, Spanish and Belgians, the early medieval (Alemannic and Bavarian) populations of southern Germany[47,48] exhibit closest resemblance to present-day Danish, northern Germans, Dutch and Scandinavians (Supplementary Fig. 5.4) and are genetically indistinguishable from Iron Age and Medieval groups in northern Germany and Scandinavia (Supplementary Table 2.10). We argue that this is the result of a major genetic influx from those regions as indicated by qpWave analysis and supervised ADMIXTURE (Fig. 4c and Supplementary Figs. 5.3, 5.5 and 5.6). The northern regions of Germany (here Saxony-Anhalt, Lower Saxony, Mecklenburg-Vorpommern and Schleswig-Holstein) underwent a very different population genetic trajectory than southern Germany. While the Bronze and Iron Age populations in the north also received additional EEF ancestry (Supplementary Figs. 4.5a,b), it was substantially less than what arrived in southern Germany, forming a Steppe ancestry-enriched gene pool highly similar to contemporaneous populations in Denmark, Sweden and Norway (Supplementary Fig. 5.2). Migration from northern Germany introduced EEF-depleted ancestry to southern Germany, resulting in a rise of the median northern European ancestry from 2.8% during the Iron Age to 62.5% during the Early Middle Ages (Supplementary Fig. 5.3), as well as in new paternal ancestry in the form of Y-chromosome haplogroups like I1-M253 (refs. 47,48). While we cannot precisely date this migration, Roman[48] and Late Iron Age[49] data from Bavaria and Thuringia indicate that parts of the early Iron Age gene pool in southern Germany were not affected until the fourth or fifth century CE (with northern European ancestry not exceeding a median of 8% in these samples). In general, this turnover seems to be part of a larger movement of people, contributing northern European ancestry to the early medieval populations of England[50], Hungary[51], Italy[51] and Spain[52].

Most present-day Germans fall between the Hallstatt and early medieval southern German clusters, suggesting a resurgence of EEF-enriched ancestry, especially in southern Germany. This is also indicated by uniparental Y-chromosome evidence. We find that the Hallstatt Y-chromosome gene pool is dominated by R1b-M269 and G2a-P303 lineages, with subhaplogroup G2a-L497 accounting for 37% of the haplotypes in the sample (Supplementary Table 1.1). Interestingly, we find that individuals with haplogroup G2a-L497 (for example, MBG017, MBG016 and HOC004) exhibit significantly more southern European ancestry than individuals carrying haplogroup R1b-M269 (for example, HOC001, APG001 and MBG003) (two-sided Welch two-sample $t$-test; $t = 2.878$, d.f. 13.812, $P = 0.0123$). Although G2a is exceedingly rare in present-day Europe north of the Alps, G2a-L497 still peaks in the area of the former West-Hallstattkreis, namely eastern France, southern Germany, and Switzerland[53] as well as northern Italy, thus providing additional evidence for a survival or resurgence of Hallstatt Iron Age ancestry in those regions. Most present-day Germans can be modelled as three-way admixture between SGermany_EIA (54.5 ± 2%), NGermany_Roman (33.8 ± 2.5%) and a third, northeastern European source (here Latvia_BA, 11.7 ± 1.2%) representing further admixture introduced after the initial admixture event, potentially connected to Slavic-speaking populations migrating into eastern Germany during the Middle Ages[54] (Supplementary Tables 4.13–14).

## Discussion

Hereditary leadership is described as one key aspect of early historically recorded complex societies around the world[16,55], but it is hard to prove through the archaeological record only. Combining uniparental and autosomal data, we were able to prove a close biological relationship between the two central princely burials of Eberdingen-Hochdorf (HOC001) and Asperg-Grafenbühl (APG001), representing two of the richest graves of European prehistory. Together with dating and osteological estimates of age at death, our pedigree modelling points to a maternal uncle–sororal nephew relationship (most likely model) or a grandfather–daughter–grandson model, suggesting that in this case institutionalized power was matrilineally inherited from the potentate (HOC001), most probably via his sister's, and less likely via his daughter's son (APG001). The first and substantially more likely of these scenarios would be congruent with (later) historical Roman accounts of avuncularism among the early Celts of the fifth or fourth century BCE[27,56]. Today, matrilineally organized societies represent only 12–17% of the world's populations[57], with the majority of societies being patrilineally organized, a pattern also evident from aDNA studies of Neolithic and Bronze Age communities in Europe[38,58,59]. Yet, global instances of prehistoric societies where hereditary leadership was passed in multigenerational matrilineal descent groups are known[60]. For Iron Age Europe, matrilineal inheritance of regality is documented for Etruria and Ancient Rome[27].

Matrilinear avunculate organization is shown to emerge in populations in which extramarital mating is common and/or paternity confidence is low, so that men are more likely genetically closer related to their sisters' children than to those of their own wives, ultimately favouring investment in sisters' children[61–66]. In this context, the observation of inbreeding in two individuals from Asperg-Grafenbühl and Magdalenenberg is indicative. Both individuals are most likely the product of first-cousin mating, a practice often associated with paternity certainty and avunculocal organization, which allows males in matrilineal societies to contribute to sisters' children who are married to their own wife's children[64,67]. In the aDNA record, first cousin mating is exceedingly rare, with less than 3% of ancient individuals showing RoH consistent (but not conclusive) for the offspring of first cousins[40]. Yet, we highlight that this leadership system may be limited to southern Germany and not apply to the rest of the Hallstatt sphere. In addition, there might be differences between the elite and the larger common population. Recent genetic evidence from the Hallstatt Dolge njive barrow cemetery in Slovenia is neither consistent with a strictly matrilineal nor patrilineal kinship structure for the buried population[68] and might indicate a more complex heritability system along both the male and female lines that potentially included adoption or fosterage as well[68].

In this context, we find no genetic relationship between the consanguineous Asperg-Grafenbühl child (APG003) and the adult female he was buried with (APG002), nor the main burial, potentially representing an instance of 'alliance fosterage'[69–71], a practice associated with the establishment of reciprocal claims on loyalty between status groups and ultimately feudatory state formation[72,73]. Additionally, a fosterage model would also be supported by his $^{87}Sr/^{86}Sr$ values, indicating that he originated from the periphery of the Magdalenenberg site, agreeing with written records of non-kinship fostering among the continental and insular Celtic elites[70,71].

We find further evidence of familial interconnectedness between the earlier site of Magdalenenberg and the later Eberdingen-Hochdorf in the form of a third-degree genetic relationship between MBG009 and HOC003 and seventh- to eighth-degree relationships between the Magdalenenberg princely burial MBG017, secondary burial MBG016 and HOC004. Together with the relationship between HOC001 and APG001, these connections link the three monumental tumuli. Such instances of non-random mating across a linear geographic distance of more than 100 km and a time span of up to 140 years suggest a high degree of social complexity and the emergence of regional-scale

hierarchy. In general, the isotopic profiles of the Magdalenenberg population indicate high, continent-wide mobility during their lifetime and may represent the signature of marital alliance structures and patronal fosterage that connected the distant elite centres and formed the far-reaching social and economic Hallstatt networks[37,39].

The early Celtic elite of these networks emerged from a long-term population genetic process of ongoing admixture with coexisting groups in southern Europe who previously experienced less gene flow from Steppe-related populations[38,42]. In this context, we highlight our finding that the earliest elite burial in the region from the central grave of the Magdalenenberg at 616 BCE, as well as his relatives, show evidence of ancestry from South of the Alps, which might suggest a leading role of this connection in the initial formation of the early Celtic Hallstatt culture. Cultural links across the Alps are also preserved in the material culture of these elite graves throughout centuries[10,12,39]. However, the complex political structures disintegrated in the fifth and fourth century BCE and were ultimately abandoned. Genetic outliers from this and previously published studies suggest that, subsequently, at the height of the Celtic migrations during the fourth and third century BCE, not only 'Celts' migrated, but at least a limited number of people from northern central Europe reached the southern zone of the La Tène culture and even northern Italy[74], possibly associated with historical entities like the Cimbri and Teutones[75]. The historical and archaeological record leave no doubt that the development of culture and population in southwestern Germany was temporarily characterized by profound discontinuities, particularly during the third to first century BCE. The definitive end of the 2,000 years of relative genetic continuity from the Bronze throughout the Iron Age in southern Germany is marked by a sudden, sharp increase of Steppe-related ancestry during the Late Antiquity and Early Middle Ages. From a population genetic perspective, this is congruent with the arrival of Germanic-speaking tribes from northern Germany or Denmark during the migration period, as also documented by inscription records in the sixth- and seventh-century sites of Baden-Württemberg and Bavaria[76]. Together with ancestry from eastern Europe introduced during the Middle Ages[54], as well as more recent genetic influx from all over the globe, those ancestral populations form the gene pool of the present-day German population.

## Methods
### aDNA sequencing
**Archaeological research.** Provenance information for samples from all archaeological sites are given in Supplementary Note 1, together with descriptions of each site, the institution owning the samples (or custodians of the samples), the responsible co-author who obtained permission to analyse, and the year of the permission granted.

**Sampling.** Sampling of 31 bone and teeth samples took place in clean-room facilities dedicated to aDNA work, for 23 samples at the Max Planck Institute for Science of Human History in Jena (MPI-SHH), for 5 at the Institute for Archaeological Sciences of the Eberhard Karls University Tübingen and for 3 at the EURAC Institute for Mummy Studies in Bolzano, Italy. The sampling workflow included documenting and photographing the provided samples. For teeth, we either cut along the cementum–enamel junction and collected powder by drilling into the pulp chamber or accessed the pulp chamber by drilling the tooth transversally. For the petrous bones, we cut the petrous pyramid longitudinally to drill the dense part directly from either side[77]. We collected between 30 and 200 mg of bone or tooth powder per sample for DNA extractions.

**DNA extraction.** aDNA was extracted following a modified protocol of Dabney et al.[78], as described in www.protocols.io/view/ancient-dna-extraction-from-skeletal-material-baksicwe, where we replaced the extended-MinElute-column assembly for manual extractions with columns from the Roche High Pure Viral Nucleic Acid

Large Volume Kit[79], and for automated extraction with a protocol that replaced spin columns with silica beads in the purification step[80].

**Library construction.** We generated 22 double-indexed[81] double-stranded libraries using 25 µl of DNA extract and following established protocols[82]. We applied the partial uracil–DNA–glycosylase treatment (UDG-half)[83] protocol to remove most of the aDNA damage while preserving the characteristic damage pattern in the terminal nucleotides. For 13 extracts, we generated double-indexed single-stranded libraries[84] using 20 µl of DNA extract and applied no uracil-DNA-glycosylase treatment.

**Shotgun screening, capture and sequencing.** Libraries were sequenced in-house on an Illumina HiSeq 4000 platform to an average depth of 5 million reads and after demultiplexing processed through EAGER[85]. After an initial quality filter based on the presence of aDNA damage and endogenous DNA higher than 0.1%, we subsequently enriched 35 libraries using in-solution capture probes synthesized by Agilent Technologies for ~1,240k SNPs along the nuclear genome[86]. The captured libraries were sequenced for ~50 million reads on average (minimum 20 million, maximum 140 million) using a single-end (1 × 75 bp reads) configuration.

### aDNA data processing
**Read processing and aDNA damage.** After demultiplexing based on a unique pair of indexes, raw sequence data were processed using EAGER[85]. This included clipping sequencing adaptors from reads with AdapterRemoval (v2.3.1)[87] and mapping of reads with BWA (Burrows–Wheeler Aligner) v0.7.12 (ref. [88]) against the Human Reference Genome hg19, with seed length (-l) disabled, maximum number of differences (-n) of 0.01 and a quality filter (-q) of 30. We removed duplicate reads with the same orientation and start and end positions using DeDup v0.12.2 (ref. [85]). Terminal base deamination damage calculation was done using mapDamage v2.0.6 (ref. [89]), specifying a length (-l) of 100 bp. For the 22 libraries that underwent UDG half treatment, we used BamUtil v1.0.14 (https://genome.sph.umich.edu/wiki/BamUtil:_trimBam) to clip two bases at the start and end of all reads for each sample to remove residual deaminations, thus removing genotyping errors that could arise due to aDNA damage.

**Sex determination.** To determine the genetic sex of the ancient individuals, we calculated the coverage on the autosomes as well as on each sex chromosome and subsequently normalized the X and Y reads by the autosomal coverage[90]. For that, we used a custom script (https://github.com/TCLamnidis/Sex.DetERRmine) for the calculation of each relative coverage as well as their associated error bars[91]. Females are expected to have an X rate of 1 and a Y rate of 0, while males are expected to have a rate of 0.5 for both X and Y chromosomes.

**Contamination estimation.** We used the ANGSD (analysis of next-generation sequencing data) package[92] (v0.923) to test for heterozygosity of polymorphic sites on the X chromosome in male individuals, applying a contamination threshold of 5% at the results of method 2. For male and female samples, we estimated contamination levels on the mtDNA using Schmutzi[93] (v1.5.4) by comparing the consensus mitogenome of the ancient sample to a panel of 197 worldwide mitogenomes as a potential contamination source, applying a contamination threshold of 5%. We used PMDtools[94] (v0.50) to isolate sequences from each sample that had clear evidence of contamination (over 5% on the X chromosome or mitogenome) according to the post-mortem damage score (PMD score >3, using only bases with phred-scaled quality of at least 30 to compute the score), and performed contamination estimation again.

**Genotyping.** We used the program pileupCaller (v1.4.0.2) (https://github.com/stschiff/sequenceTools.git) to genotype the trimmed BAM files of 22 UDG half libraries. A pileup file was generated using samtools

mpileup with parameters -q 30 -Q 30 -B containing only sites overlapping with our capture panel. From this file, for each individual and each SNP on the 1,240k panel[95–97], one read covering the SNP was drawn at random and a pseudo-haploid call was made, that is, the ancient individual was assumed homozygous for the allele on the randomly drawn read for the SNP in question. For the 13 single-stranded libraries that underwent no UDG treatment, we used the parameter -SingleStrandMode, which causes pileupCaller to ignore reads aligning to the forward strand at C/T polymorphisms and at G/A polymorphisms to ignore reads aligning to the reverse strand, which should remove post-mortem damage in aDNA libraries prepared with the non-UDG single-stranded protocol.

**Mitochondrial and Y-chromosome haplogroup assignment.** To process the mitochondrial DNA data, we extracted reads from 1,240k data using samtools (v1.3.1)[98] and mapped these to the revised Cambridge reference sequence. We subsequently called consensus sequences using Geneious R9.8.1 (ref. [99]) and used HaploGrep 2 (v2.4.0)[100] (https://haplogrep.uibk.ac.at/; with PhyloTree version 17-FU1) to determine mitochondrial haplotypes. For the male individuals, we used pileup from the Rsamtools package to call the Y-chromosome SNPs of the 1,240k SNP panel (mapping quality ≥30 and base quality ≥30). We then manually assigned Y-chromosome haplogroups using pileups of Y-SNPs included in the 1,240k panel that overlap with SNPs included on the ISOGG SNP index v.15.73 (Y-DNA Haplogroup Tree 2019-2020; 2020.07.11).

**Kinship estimation.** We calculated the pairwise mismatch rate[60] in all pairs of individuals from our pseudo-haploid dataset to double-check for potential duplicate individuals and to determine first-, second- and third-degree relatives. For this purpose, we also used BREADR[101] which utilizes Bayesian posterior probabilities for the classification of the genetic relationships. Additionally, we also applied LcMLkin[102] (v0.5.0) and KIN[103] (v3.1.3), which use genotype likelihoods to estimate the three $k$ coefficients ($k_0$, $k_1$ or $k_2$), which define the probability that two individuals have zero, one or two alleles identical by descent at a random site in the genome (Supplementary Note 2).

**Inbreeding estimation.** We calculated the length of RoH using the software HapROH (v0.6)[40]. An SNP cut-off of 300,000 SNPs was used, as well as the default 1000 Genomes reference panel.

**IBD.** We imputed and phased individuals with more than 390,000 SNPs using GLIMPSE[104] (v2.0.0) (https://github.com/odelaneau/GLIMPSE), applying the default parameters and using the 1000 Genomes reference panel. Samples with more than 600k SNPs exhibiting a genotype posterior of ≥0.99 after imputation were included in downstream IBD analysis. We used ancIBD[31] (v0.4) (https://pypi.org/project/ancIBD/) to call and summarize IBD blocks of 8, 12, 16 and 20 cM size shared between pairs of individuals.

**Latent pedigree modelling.** Details are described in Supplementary Note 3. We investigated a total of 11 plausible pedigrees connecting the Hochdorf and Asperg central burials, compatible with either first- or second-degree relatedness. We modelled the likelihood of each pedigree based on observed data (date ranges of their burials, estimates of their age-at-death), and parameterised with unknowns, such as mother's ages at the birth of both individuals and of other key pedigree members. We computed the joint posterior probability for the parameters of the model using Markov Chain Monte Carlo sampling. We then computed marginal likelihoods for each model based on the posterior samples, and combined these with the respective probabilities for genetic kinship of each model, as well as the probability for matching mitochondrial sequences. Taken all probabilities together, emerging from combining archaeological data (dating), anthropological data (age at death, and priors for mother's ages) and genetic data (autosomal kinship and the probability to observe a matching mitochondrial

sequence), we produced a final ranking of 11 models, each with a combined marginal likelihood and Bayes factors.

## Population genetic analysis

**Dataset.** We merged our aDNA data with previously published datasets of 5,665 ancient individuals reported by the Reich Lab in the Allen Ancient DNA Resource v54.1 (https://reich.hms.harvard.edu/allen-ancient-dna-resource-aadr-downloadable-genotypes-present-day-and-ancient-dna-data). We assembled a dataset from mostly European populations for genome-wide analyses[105–115]. This modern set includes 10,176 individuals. Loci and individuals with <95% call rate as well as a 15 Mb region surrounding the HLA region[115] were removed and loci on three previously reported long range LD regions on chromosomes 6, 8 and 11 (refs. [116],[117]) were pruned using PLINK[118] (v1.90b3.29). aDNA data were merged to this dataset, correcting for reference allele and strand flips. We kept 445,171 autosomal SNPs after intersecting autosomal SNPs in the 1,240k capture with the modern analysis set.

**Abbreviations.** We have used the following abbreviations in population labels: N, Neolithic; C, Chalcolithic; EBA, Early Bronze Age; MBA, Middle Bronze Age; LBA, Late Bronze Age; Iron Age, IA; RA, Roman Age; EMA, Early Middle Ages; MA, Middle Ages. In Germany, these periods roughly correspond to the following simplified time ranges: Neolithic: 4000 to 2500 BCE, Chalcolithic and EBA: 2500 to 1600 BCE; MBA: 1600 to 1200 BCE; LBA: 1200 to 800 BCE; IA: 800 BCE to 400 CE; EMA 400 to 1000 CE.

**PCA.** We carried out PCA using the smartpca software v16000 from the EIGENSOFT package (v6.0.1)[119]. We computed principal components on two different sets of modern European populations (Supplementary Note 4) as well as on 59 West Eurasian groups (following Lazaridis et al.[97]) and projected ancient individuals using lsqproject: YES. We used the PCA output for MOBEST[35] analysis as described by the authors (https://github.com/nevrome/mobest). PCA on the Steppe, WHG and EEF components measured in 153 ancient and present-day populations was calculated using the prcomp function from the stats package (v3.6.2) in R (v4.1.1).

**F statistics.** $F_3$ and $F_4$ statistics were computed with ADMIXTOOLS v3.0 (ref. [120]) (https://github.com/DReichLab). $F_3$ statistics were calculated using qp3Pop (v435). For $F_4$ statistics, we used the qpDstat (v755) and with the activated $F_4$ mode. Significant deviation from zero can be interpreted as rejection of the tree population typology ((Outgroup, X);(Pop1, Pop2)). Under the assumption that no gene flow occurred between Pop1 and Pop2 and the Outgroup, a positive $F_4$ $f$ statistic suggests affinity between X and Pop2, while a negative value indicates affinity between X and Pop1. Standard errors were calculated with the default block jackknife 5 cM in size. As outgroups we used either Mbuti.DG, YRI.SG or CHB.SG.

**Fixation index.** We calculated $F_{ST}$ using smartpca software v16000 from the EIGENSOFT package (v6.0.1)[119] with the fstonly, inbreed and fsthiprecision options set to YES.

**Inference of mixture proportions and sex bias.** We estimated ancestry proportions using qpWave[95,121] (v410) and qpAdm[95] (v810) from ADMIXTOOLS v3.0 (ref. [120]) with the allsnps: YES option and two basic sets of 11 (ref. [122]) (for qpWave analysis) and 4 (ref. [42]) (for distal qpAdm analysis) outgroups, respectively:

I. YRI.SG, Poland, Finland, Sweden, Denmark, Ireland, Wales, Italy, Spain, Belgium and the Netherlands.
II. OldAfrica, WHGB and Turkey_N, Afanasievo.

To analyse potential sex bias in the admixture process, we used qpAdm to estimate EEF admixture proportions on the autosomes (default option) and on the X chromosome (option 'chrom: 23') using

the left and right populations described in Patterson et al.[42]. Following the approach established by Mathieson et al. (2018), $Z$ scores were calculated for the difference between the autosomes and the X chromosome using the formula $Z = \frac{p_A - p_X}{\sqrt{\sigma_A^2 + \sigma_X^2}}$ where $p_A$ and $p_X$ are the EEF admixture proportions on the autosomes and the X chromosome, and $\sigma_A$ and $\sigma_X$ are the corresponding jackknife standard deviations[36]. Thus, a negative $Z$ score means that there is more EEF admixture on the X chromosome than on the autosomes, indicating that the EEF admixture was female biased.

**ADMIXTURE analysis.** We performed model-based clustering analysis using ADMIXTURE[123] (v1.3). We used ADMIXTURE in supervised mode, where we estimated admixture proportions for the ancient individuals using modern reference populations at $K = 12$. Following the approach described in Gretzinger and colleagues[122] and Supplementary Note 4, these analyses were run on haploid data with the parameter –haploid set to all (='*'). Standard errors for point estimates were calculated using 1,000 bootstrap replicates with the -B parameter. To obtain point estimates for populations, we averaged individual point estimates and calculated the standard error of the mean (s.e.m.) as $\frac{\sigma}{\sqrt{n}}$. We find that this better reflects the diversity within the population than a propagation of error approach, which underestimates the variance within the point estimate sample.

**Admixture dating.** Admixture dates between Steppe and EEF sources were calculated using DATES (distribution of ancestry tracts of evolutionary signals) (v4010)[124] using default settings.

### Isotope analysis

We measured strontium and oxygen isotope compositions in 17 individuals who were not previously analysed in Oelze et al.[37]. Isotope analysis was conducted at the Curt-Engelhorn-Center Archaeometry gGmbH, Mannheim, Germany. Sample preparation and analyses of strontium and oxygen isotope compositions followed previously described steps[28,125,126]. Enamel fragments were cut from the crowns using a diamond-coated cutting disc attached to a dental drill. All surfaces and remaining dentin were removed using diamond-coated milling bits and the samples powdered in an agate mortar. For Sr isotope analysis, 11–12 mg of sample material were pre-treated to remove diagenetic carbonates. In successive steps, the powder was placed in an ultrasonic bath for 10 min each with 1.8 ml of supra-pure $H_2O$ and 1.8 ml of 0.1 M acetic acid buffered with lithium acetate (pH ca. 4.5) and three times with 1.8 ml of $H_2O$. Samples were afterwards dried overnight (50 °C) and ashed to remove remaining organic components (3 h at 850 °C). All subsequent steps were carried out under clean lab conditions. The samples were dissolved in nitric acid (3 N $HNO_3$), and the strontium was separated using Sr-Spec ion exchange resin. Strontium concentrations were determined using an optical emission spectrometry with inductively coupled plasma ionization (ICP-OES iCAP 7200), the solutions diluted and the isotope ratios determined using a high-resolution multi-collector inductively coupled plasma mass spectrometer (Neptune). The raw data were corrected according to the exponential mass fractionation law to $^{88}Sr/^{86}Sr = 8.375209$. Blank values were less than 10 pg Sr during the clean lab procedure, including digestion, Sr separation and measurement. Standards run with the samples produced the following values:

| Standard | Number | $^{87}Sr/$ $^{86}Sr$ Avg | 2 Sigma | Certified value/ interlaboratory mean | Reference |
|---|---|---|---|---|---|
| NBS-987 | 10 | 0.71030 | 0.00002 | 0.71034± 0.00026 (95% confidence interval) | https://www-s. nist.gov/srmors/ certificates/987 .pdf |
| NBS-987 after Sr separation | 2 | 0.71030 | 0.00002 | 0.71034± 0.00026 (95% confidence interval) | https://www-s. nist.gov/srmors/ certificates/987 .pdf |

In this study, we determined the isotope composition of the oxygen bound in the phosphate component of the hydroxyapatite. Ten milligrams of the enamel powder of each tooth were pre-treated with 1.8 ml of 2.5% NaOCl for 24 h, rinsed three times with supra-pure water, reacted in 800 μl of 2 M HF overnight, shaken and centrifuged, and the solutions were transferred into new sample tubes, leaving the CaF residues behind[28,126]. After adding ca. 200 μl of bromothymol blue indicator, the HF was neutralized with ca. 140 μl of 25% $NH_4OH$ solution. The addition of 800 μl of 2 M $AgNO_3$ solution caused the phosphate ions to precipitate immediately as $Ag_3PO_4$, which was washed five times and dried overnight at 50 °C. The samples were analysed in triplicates. Pyrolysis was performed using a vario PYRO cube CNSOH elemental analyser (Elementar). For isotope analysis, the resulting CO was transferred into a precisION isotope ratio mass spectrometer (Isoprime). Raw data were corrected against IVA silver phosphate ($Ag_3PO_4$) with $\delta^{18}O = 21.7‰$ (certificate no. BN 180097) using the internal software (single-point-normation). Three kinds of standard materials were prepared and analysed along with the samples: NBS 120c gave $\delta^{18}O$ values of $22.00 \pm 0.26‰$ ($n = 6$). The in-house standards of synthetic hydroxyapatite gave $17.23 \pm 0.22‰$ ($n = 6$) and Roman pig bones from the site of Dangstetten (SUS-DAN) gave $14.68 \pm 0.21‰$ ($n = 6$).

### Reporting summary

Further information on research design is available in the Nature Portfolio Reporting Summary linked to this article.

### Data availability

Raw sequence data (fastq files) and mapped data (bam files) from the 31 newly reported ancient individuals will be available prior publication from the European Nucleotide Archive under accession number PRJEB73566. Published genotype data for the present-day British sample are available from the WTCCC via the European Genotype Archive (https://www.ebi.ac.uk/ega/) under accession number EGAD00010000634. Published genotype data for the present-day Irish sample are available from the WTCCC via the European Genotype Archive under accession number EGAD00010000124. Published genotype data for the rest of the present-day European samples are available from the WTCCC via the European Genotype Archive under accession number EGAD00000000120. Published genotype data for the Dutch samples are available by the GoNL request process from The Genome of the Netherlands Data Access Committee (DAC) (https://www.nlgenome.nl). The Genome Reference Consortium Human Build 37 (GRCh37) is available via the National Center for Biotechnology Information under accession number PRJNA31257. The revised Cambridge reference sequence is available via the National Center for Biotechnology Information under NCBI Reference Sequence NC_012920.1. Previously published genotype data for ancient individuals were reported by the Reich Lab in the Allen Ancient DNA Resource v.54.1(https://reich.hms.harvard.edu/allen-ancient-dna-resource-aadr-downloadable-genotypes-present-day-and-ancient-dna-data), as well as Poseidon (https://www.poseidon-adna.org). A Poseidon package of the genotype data analysed in this paper is available on the Poseidon Community Archive (https://www.poseidon-adna.org/#/archive_explorer).

### Code availability

All software used in this work is publicly available. Custom code developed for the Bayesian pedigree modelling (Supplementary Text 3) is described and available at https://github.com/stschiff/celtic_relationship_analysis. An archived version is available on Zenodo (10.5281/zenodo.10427675). Corresponding publications are cited in the main text and supplementary material. List of software and respective versions: AdapterRemoval (v2.3.1), Burrows–Wheeler Aligner (v0.7.12), DeDup (v0.12.2), mapDamage (v2.0.6), BamUtil (v1.0.14), EAGER (v1), Sex.DetERRmine (v1.1.2) (https://github.com/

TCLamnidis/Sex.DetERRmine), ANGSD (v0.923), Schmutzi (v1.5.4), PMDtools (v0.50), pileupCaller (v1.4.0.2), samtools (v1.3.1), Geneious R9.8.1, HaploGrep 2 (v2.4.0), READ (https://bitbucket.org/tguenther/read) (vf541d55), lcMLkin (https://github.com/COMBINE-lab/maximum-likelihood-relatedness-estimation) (v0.5.0), PLINK (v1.90b3.29), Picard tools (v2.27.3), smartpca (v16000; EIGENSOFT v6.0.1), qp3Pop (v.435; ADMIXTOOLS v3.0), qpDstat (v.755; ADMIXTOOLS v3.0), qpWave (v410), qpAdm (v.810), hapROH (v0.6), DATES (v4010), ADMIXTURE (v1.3), KIN (v3.1.3), ancIBD (https://pypi.org/project/ancIBD) (v0.4), GLIMPSE (https://github.com/odelaneau/GLIMPSE) (v2.0.0), BREADR (https://github.com/jonotuke/BREADR) (746316f), MOBEST (https://github.com/nevrome/mobest.analysis.2022) (v26f929e), MAFFT (v6.864) and MEGA (v7). Data visualization and descriptive statistical tests were performed in R (v4.1.1). The following R packages were used: Rsamtools (v2.12.0), binom (v1.1-1.1), ape (v.5.6-2), phytools (v1.0-3), psych (v2.2.5), vegan (v2.6-2), factoextra (v1.0.7), ggplot2 (v3.3.6), ggExtra (v0.10.0), ggforce (v0.3.3), rnaturalearth (v0.1.0), sf (v1.0-.8), raster (v3.5-21), elevatr (v0.4.2), rgdal (v1.5-32), spatstat (v2.3-4), maptools (v1.1-4), gstat (v2.0-9), sp (v1.5-0), labdsv (v2.0-1), igraph (v1.3.4), magrittr (v2.0.3), dplyr (v1.0.9), reshape 2 (v1.4.4) and tidyverse (v.1.3.2). Y-chromosome and mtDNA haplogroups were determined using the ISOGG SNP index (v15.73) and PhyloTree (v17-FU1) reference databases, respectively.

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

## Acknowledgements

This project has received funding from the European Research Council (ERC) under the European Union's Horizon 2020 research and innovation programme (grant agreement number 851511) (S.S.). K.H. is supported by the DFG (FOR 2237) and the ERC (grant agreement number 101019659). Mila Sproß, Sigrid Klaus and Bernd Höppner contributed to isotope analyses at the Curt-Engelhorn-Center

Archaeometry gGmbH, Mannheim, Germany. The funders had no role in study design, data collection and analysis, decision to publish or preparation of the manuscript.

## Author contributions

W.S., D.K. and J.K. conceived the study. F.S., M.F., L.H., H. Ra, K.H., G.W., F. Man and D.K. provided archaeological samples and context. J.G., C.F., C.P., L.G. and F. Mai performed laboratory analyses. J.G., A.M., S.C., T.L., Y.H., H. Ri, C.K. and S.S. analysed data. A.Z., W.S., D.K., J.K. and S.S. supervised the project. J.G., A.M., D.K. and S.S. wrote the paper with input from all co-authors.

## Funding

## Competing interests

The authors declare no competing interests.

## Additional information

**Correspondence and requests for materials** should be addressed to Dirk Krausse, Johannes Krause or Stephan Schiffels.

# Reporting Summary

## Statistics

For all statistical analyses, confirm that the following items are present in the figure legend, table legend, main text, or Methods section.

| n/a | Confirmed | |
|---|---|---|
| ☐ | ☒ | The exact sample size (*n*) for each experimental group/condition, given as a discrete number and unit of measurement |
| ☒ | ☐ | A statement on whether measurements were taken from distinct samples or whether the same sample was measured repeatedly |
| ☐ | ☒ | The statistical test(s) used AND whether they are one- or two-sided *Only common tests should be described solely by name; describe more complex techniques in the Methods section.* |
| ☒ | ☐ | A description of all covariates tested |
| ☐ | ☒ | A description of any assumptions or corrections, such as tests of normality and adjustment for multiple comparisons |
| ☐ | ☒ | A full description of the statistical parameters including central tendency (e.g. means) or other basic estimates (e.g. regression coefficient) AND variation (e.g. standard deviation) or associated estimates of uncertainty (e.g. confidence intervals) |
| ☐ | ☒ | For null hypothesis testing, the test statistic (e.g. *F*, *t*, *r*) with confidence intervals, effect sizes, degrees of freedom and *P* value noted *Give P values as exact values whenever suitable.* |
| ☐ | ☒ | For Bayesian analysis, information on the choice of priors and Markov chain Monte Carlo settings |
| ☒ | ☐ | For hierarchical and complex designs, identification of the appropriate level for tests and full reporting of outcomes |
| ☐ | ☒ | Estimates of effect sizes (e.g. Cohen's *d*, Pearson's *r*), indicating how they were calculated |

*Our web collection on statistics for biologists contains articles on many of the points above.*

## Software and code

Policy information about availability of computer code

| Data collection | No specific software was used for sample collection. Genotype data were generated from sequencing reads. All software used in this study is listed below. |
|---|---|
| Data analysis | All software used in this work is publicly available. Custom code developed for the Bayesian pedigree modelling (Supplementary Text 3) is described and available at https://github.com/stschiff/celtic_relationship_analysis. An archived version is available on zenodo (10.5281/zenodo.10427675). Corresponding publications are cited in the main text and supplementary material. List of software and respective versions: AdapterRemoval (v2.3.1), Burrows-Wheeler Aligner (v0.7.12), DeDup (v0.12.2), mapDamage (v2.0.6), BamUtil (v1.0.14), EAGER (v1), Sex.DetERRmine (v1.1.2) (https://github.com/TCLamnidis/Sex.DetERRmine), ANGSD (v0.923), Schmutzi (v1.5.4), PMDtools (v0.50), pileupCaller (v1.4.0.2), samtools (v1.3.1), Geneious R9.8.1, HaploGrep 2 (v2.4.0), READ (https://bitbucket.org/tguenther/read) (vf541d55), lcMLkin (https://github.com/COMBINE-lab/maximum-likelihood-relatedness-estimation) (v0.5.0), PLINK (v1.90b3.29), Picard tools (v2.27.3), smartpca (v16000; EIGENSOFT v6.0.1), qp3Pop (v.435; ADMIXTOOLS v3.0), qpDstat (v.755; ADMIXTOOLS v3.0), qpWave (v410), qpAdm (v.810), hapROH (v0.6), DATES (v4010), ADMIXTURE (v1.3), KIN (v3.1.3), ancIBD (https://pypi.org/project/ancIBD) (v0.4), GLIMPSE (https://github.com/odelaneau/GLIMPSE) (v2.0.0), BREADR (https://github.com/jonotuke/BREADR) (746316f), MOBEST (https://github.com/nevrome/mobest.analysis.2022) (v26f929e), MAFFT (v6.864), MEGA (v7). Data visualisation and descriptive statistical tests were performed in R (v4.1.1). The following R packages were used: Rsamtools (v2.12.0), binom (v1.1-1.1), ape (v.5.6-2), phytools (v1.0-3), psych (v2.2.5), vegan (v2.6-2), factoextra (v1.0.7), ggplot2 (v3.3.6), ggExtra (v0.10.0), ggforce (v0.3.3), rnaturalearth (v0.1.0), sf (v1.0.-8), raster (v3.5-21), elevatr (v0.4.2), rgdal (v1.5-32), spatstat (v2.3-4), maptools (v1.1-4), gstat (v2.0-9), sp (v1.5-0), labdsv (v2.0-1), igraph (v1.3.4), magrittr (v2.0.3), dplyr (v1.0.9), reshape 2 (v1.4.4), and tidyverse (v.1.3.2). Y-chromosome and mtDNA haplogroups were determined using the ISOGG SNP index (v15.73) and PhyloTree (v17-FU1) reference databases, respectively. |

For manuscripts utilizing custom algorithms or software that are central to the research but not yet described in published literature, software must be made available to editors and reviewers. We strongly encourage code deposition in a community repository (e.g. GitHub). See the Nature Portfolio guidelines for submitting code & software for further information.

## Data

Policy information about availability of data

All manuscripts must include a data availability statement. This statement should provide the following information, where applicable:

- Accession codes, unique identifiers, or web links for publicly available datasets
- A description of any restrictions on data availability
- For clinical datasets or third party data, please ensure that the statement adheres to our policy

Raw sequence data (fastq files) from the 31 newly reported ancient individuals will be available prior publication from the European Nucleotide Archive under accession number PRJEB73566. Published genotype data for the present-day British sample are available from the WTCCC via the European Genotype Archive (https://www.ebi.ac.uk/ega/) under accession number EGAD00010000634. Published genotype data for the present-day Irish sample are available from the WTCCC via the European Genotype Archive under accession number EGAD00010000124. Published genotype data for the rest of the present-day European samples are available from the WTCCC via the European Genotype Archive under accession number EGAD00000000120. Published genotype data for the Dutch samples are available by the GoNL request process from The Genome of the Netherlands Data Access Committee (DAC) (https://www.nlgenome.nl). The Genome Reference Consortium Human Build 37 (GRCh37) is available via the National Center for Biotechnology Information under accession number PRJNA31257. The revised Cambridge reference sequence is available via the National Center for Biotechnology Information under NCBI Reference Sequence NC_012920.1. Previous published genotype data for ancient individuals was reported by the Reich Lab in the Allen Ancient DNA Resource v.54.1(https://reich.hms.harvard.edu/allen-ancient-dna-resource-aadr-downloadable-genotypes-present-day-and-ancient-dna-data).

## Research involving human participants, their data, or biological material

Policy information about studies with human participants or human data. See also policy information about sex, gender (identity/presentation), and sexual orientation and race, ethnicity and racism.

| | |
|---|---|
| Reporting on sex and gender | Does not apply. This study does not include novel data from present-day humans or any present-day human participants. |
| Reporting on race, ethnicity, or other socially relevant groupings | Does not apply. This study does not include novel data from present-day humans or any present-day human participants. |
| Population characteristics | Does not apply. This study does not include novel data from present-day humans or any present-day human participants. |
| Recruitment | Does not apply. This study does not include novel data from present-day humans or any present-day human participants. |
| Ethics oversight | Does not apply. This study does not include novel data from present-day humans or any present-day human participants. |

Note that full information on the approval of the study protocol must also be provided in the manuscript.

# Field-specific reporting

Please select the one below that is the best fit for your research. If you are not sure, read the appropriate sections before making your selection.

☒ Life sciences ☐ Behavioural & social sciences ☐ Ecological, evolutionary & environmental sciences

For a reference copy of the document with all sections, see nature.com/documents/nr-reporting-summary-flat.pdf

# Life sciences study design

All studies must disclose on these points even when the disclosure is negative.

| | |
|---|---|
| Sample size | We did not rely on statistical methods to predetermine sample sizes. Sample sizes for ancient populations depended solely on the availability of archaeological material and on ancient DNA preservation. For present-day populations, sample sizes are predefined by the availability of published data. In our study, most present-day populations are represented by more than 100 genomes. The selection of samples and sample size calculation for these data is described in the source publications and follows the established guidelines in medical genetics. For both ancient and present-day populations, we aim to maximizes sample sizes by including all genomes that fulfil our quality criteria mentioned in the Methods section and the quality criteria described in their respective source publications. The reported standard errors used to describe uncertainty ranges of our statistical analyses often reflect both sample size and data quality per sample. |
| Data exclusions | One sample (SCN001) was excluded from genome-wide analyses since the authenticity of the autosomal ancient DNA data could not be ensured. The quality criteria forming the basis of this decision are mentioned in the Methods section are pre-established by various previous publications. |
| Replication | We studied unique entities (past and present populations) and did not perform experiments or study various treatments, so replication is not applicable. But we note that samples from the same population carry similar genetic signatures. For the four samples HOC001, APG001, APG003 and MBG009, we produced complementary to the partial UDG-treated, double-stranded DNA libraries also non UDG-treated single-stranded libraries to increase the genome-wide coverage. While the proportion of authentic ancient DNA obtained from the single-stranded libraries is generally comparable to the proportion measured in the double-stranded libraries, the combination of genotypes from both sources substantially increases the coverage of the respective genomes. Testing of pairwise mismatches in 1,24 Million SNP sites between two |

libraries of the same individual confirm that DNA sequences from both double- and single-stranded libraries are indeed identical. Also, genome-wide data allows for the analysis of multiple realisations of the sample history, by studying hundreds of thousands of SNP sites.

| | |
|---|---|
| Randomization | We studied unique entities (past and present populations) and did not perform experiments or study various treatments, so randomization is not applicable. However, many of our analyses based on f-statistics involve a block jackknife to obtain uncertainty ranges through analysis of uncorrelated segments of the genome. |
| Blinding | We studied unique entities (past and present populations) and did not perform experiments or study various treatments, so blinding is not applicable. |

# Reporting for specific materials, systems and methods

We require information from authors about some types of materials, experimental systems and methods used in many studies. Here, indicate whether each material, system or method listed is relevant to your study. If you are not sure if a list item applies to your research, read the appropriate section before selecting a response.

## Materials & experimental systems

| n/a | Involved in the study |
|---|---|
| ☒ | Antibodies |
| ☒ | Eukaryotic cell lines |
| ☐ | ☒ Palaeontology and archaeology |
| ☒ | Animals and other organisms |
| ☒ | Clinical data |
| ☒ | Dual use research of concern |
| ☒ | Plants |

## Methods

| n/a | Involved in the study |
|---|---|
| ☒ | ChIP-seq |
| ☒ | Flow cytometry |
| ☒ | MRI-based neuroimaging |

## Palaeontology and Archaeology

| | |
|---|---|
| Specimen provenance | All sampled bone material belongs to the Landesamt für Denkmalpflege Baden-Württemberg, an institution of the Federal Republic of Germany. This donating partner institution is represented in the author list. Permits for destructive analyses were given explicitly |
| Specimen deposition | Specimens were returned to the owning institutions after laboratory analyses. |
| Dating methods | No new dates are provided. |

☐ Tick this box to confirm that the raw and calibrated dates are available in the paper or in Supplementary Information.

| | |
|---|---|
| Ethics oversight | No ethical oversight was required strictly. However, we confirm that all analyses followed established ethical guidelines for archaeogenetic research, as detailed in Wagner et al., AJHG, 2020 and Alpaslan-Roodenberg, Nature, 2021. |

Note that full information on the approval of the study protocol must also be provided in the manuscript.

