## [Peer Review File · Nature Human Behaviour]

Peer Review Information

Journal: Nature Human Behaviour

Manuscript Title: Evidence for dynastic succession among early Celtic elites in Central Europe

Corresponding author name(s): Dirk Krausse, Johannes Krause, Stephan Schiffels

Reviewer Comments & Decisions:

Decision Letter, initial version:

20th October 2023

Dear Dr Schiffels,

Thank you once again for your manuscript, entitled "Evidence for dynastic succession among early Celtic elites," and for your patience during the peer review process.

Your manuscript has now been evaluated by 2 reviewers, whose comments are included at the end of this letter. Although the reviewers find your work to be of interest, they also raise some important concerns. We are very interested in the possibility of publishing your study in Nature Human Behaviour, but would like to consider your response to these concerns in the form of a revised manuscript before we make a decision on publication.

When revising your work in response to reviewer comments, please address in full both of our reviewers' concerns with the pedigree analysis, and ensure that you integrate archaeological and isotopic data more thoroughly throughout the text.

In addition, your revised manuscript must comply fully with our editorial policies and formatting requirements. Failure to do so will result in your manuscript being returned to you, which will delay its consideration. To assist you in this process, I have attached a checklist that lists all of our requirements. If you have any questions about any of our policies or formatting, please don't hesitate to contact me.

In sum, we invite you to revise your manuscript taking into account all reviewer and editor comments. We are committed to providing a fair and constructive peer-review process. Do not hesitate to contact us if there are specific requests from the reviewers that you believe are technically impossible or unlikely to yield a meaningful outcome.

We hope to receive your revised manuscript within two months. I would be grateful if you could contact us as soon as possible if you foresee difficulties with meeting this target resubmission date.

- Include a "Response to the editors and reviewers" document detailing, point-by-point, how you addressed each editor and referee comment. If no action was taken to address a point, you must provide a compelling argument. When formatting this document, please respond to each reviewer comment individually, including the full text of the reviewer comment verbatim followed by your response to the individual point. This response will be used by the editors to evaluate your revision and sent back to the reviewers along with the revised manuscript.
- Highlight all changes made to your manuscript or provide us with a version that tracks changes.

[REDACTED]

We look forward to seeing the revised manuscript and thank you for the opportunity to review your work. Please do not hesitate to contact me if you have any questions or would like to discuss these revisions further.

Sincerely,

[REDACTED]

Reviewer expertise:

Reviewer #1: archaeogenetics, aDNA, prehistoric Europe

Reviewer #2: archaeology of kinship in Iron Age Celts

REVIEWER COMMENTS:

Reviewer #1:

Remarks to the Author:

The manuscript NATHUMBEHAV-23082609 by Gretzinger and collaborators entitled "Evidence for dynastic succession among early Celtic elites" provides new ancient wide-genome and isotopic data for

31 individuals from Iron Age monumental sites encompassing princely burials in the modern-day southern Germany. These famous sites have been extensively studied in an archaeological point of view and this study provides new multi-proxy results to interpret the interconnectivity of the sites, and therefore of the groups, through genetic relationship as far as 100 km apart, providing insight into matrilineal dynastic systems. This is a great study showing how the development of ancient DNA methods can provide new insights for the understanding of famous sites, really well known archaeologically.

Despite some minor comments and questions that I list below, I enthusiastically support this paper for publication, and I believe Nature Human Behaviour journal is a great support for such studies and I wish more will follow in the future.

Main point:

My main remark about the submitted paper is the lack of integration of the multi-proxy data. Ancient DNA data are really well used, whereas isotopic data, although also produced as part of this work, are forgotten at the beginning (no mention in the abstract and in the first part of the results). On another hand, archaeological data are also a bit missing in the interpretation and discussion: if the main elements are used (like southern connections in the artefacts found in the site), there is no proper discussion linking genomics, isotopes, and archaeology at the individual level. For instance, are there any artefact linking the related individuals? Are the individuals with a specific ancestry buried with specific artefacts that would link them to their area of origin (for instance LAN001)? Do the individuals sharing different ancestries also share similar artefacts? If so, is consistent with a specific geographic location? If not, does it mean the general network is culturally too homogenised? I think the paper would gain from a more integrated discussion with all the elements available, which are numerous and of great quality.

Minor points:

Main text

Introduction

- As mentioned before, it should appear in the abstract that new data are also available for isotopic analysis.
- In the introduction, I think some details about the actual chronological relation between both periods Hallstatt and La Tène are missing, while both cultures are discussed and used later in the population genetic section.
- The expression "historical sources" at the beginning of the introduction is confusing: do the authors mean actual sources from history (like Caesar maybe?) but then the reference is missing, or it's a source from an historian, as the cited references suggest it, but then it should be rephrased.
- The sentence starting with "While there are other forms of kinship, ..." is, in my opinion, incorrect, or maybe not well phrased. Dynastic system can also work through adopted children, for instance in the

neighbouring Roman Empire few centuries later. I agree that we will probably never be able to demonstrate it with our tools, especially genomics which obviously will always show biological kinship, but the sentence reads like if dynastic systems can only work with blood lines, which is not true. I would rephrase this.

Results

- There is no mention of the new isotopic results generated for this study, as stated further. They need to appear somewhere, especially as they are used later in the result section.
- To help the reader to follow the different explanations about the relationships between the individuals, I suggest to systematically provide their sex and age when there are mentioned for the first time. For instance, it's not specified for HOC003, while it is for the female MBG009. It would help to follow the text better.
- At the end of the fourth paragraph, the mention of the grave of MBG005 and the hypothesis provided is quite interesting, and would benefit of a more detailed explanation: would that case be a sacrifice? of a slave? Or could it be a voluntary death (like the sati in India)? What do the texts say about this? And how common was it? Is this grave the only example found in the sites under study?
- In the fifth paragraph, I would refer to the supplementary more precisely, as most of it becomes clear and understandable once we've read the Supplementary Notes 2 and 4. Also, I don't understand why the isotopic results are not discussed here, as some individuals show signal from Italy and Iberia...
- In the last paragraph of the results, out of the 31 individuals, and given the overall ROH values for the group, I would not call "common" the two cases of consanguinity detected here.

Emergence and decline of the West-Hallstatt gene pool

- The third paragraph shows a nice example of integrated data giving a consistent lifetime story, that's very interesting. I would suggest splitting the second sentence a bit, to make it easier to digest...
- Figure 4. Several problematic points here. The dates are given in calBP, when all the text is written in BCE. Please make it consistent, preferably in BCE, as it's the way it is used in historical times. Panel 4c is hard to understand, many elements are unclear or not explained at all: What are the pie charts representing? Is it all Germany? They need some legend or title. The sources need to be detailed somewhere. The figure needs to be understandable with going to the Supplementary, where the definition of the acronyms is actually given. If some make sense (like NOR), some really do not. What are the symbols in blue and yellow? They are confusing with the symbols given to the sites. This must be explained in the caption.

Discussion

- At the end, we would expect a deeper discussion: the paper is about kinship and population structures within a given territory. Do we see any changes in the archaeological data and sites consistent with this increase of Steppe ancestry that seems to be linked with the arrival of the

Germanic-speaking tribes? What about these princely burials? These regional structures? And what about the articulation between the Iron Age cultures Hallstatt and La Tène and the biological signals? More discussion is needed here.

Methods

- In the contamination estimation section, the second sentence is incomplete. Did you use Schmützi only? Or a second method to estimate contamination on the mitochondrial DNA?
- Where has the isotope analysis been performed?

Supplementary Information

Some typos have been detected all along the supplementary document, I suggest to carefully double-check it (for instance misspelling of La Tène, found as Latène or La Tène, or the article a/an improperly used). Also it will be necessary to double-check the layout.

Supplementary Note 1: Site descriptions

The figures are not properly named in the text and are not called when necessary (for instance p12). All this needs to be double-checked.

Supplementary Note 2: Kinship, Inbreeding and individual ancestry Kinship

- typo at the third line: "als"
- typo in paragraph 3: "Again."
- Do the authors have any explanation why the coefficient of relatedness for APG001 and HOC001 is higher than expected for a second-degree relationship? Could it be linked to a background relatedness?
- Figure 2.1: The symbol and colour for HOC003 are wrong.
- Figure 2.3: In the legend, it should be specified that these plots are built following the binomial distribution test from BREADR.

Inbreeding

- It might be useful to repeat here the threshold of 300,000 SNPs used in hapROH.
- Figure 2.7: APG003 and MBG004 are both interpreted as first-cousin's offspring, while the plots b and d show very different patterns. The plot b shows a scenario where the inbreeding seems to be consistent with one between closer parents, like full siblings. Could the authors comment on these results and their interpretations?

Individual ancestry

- Tables 2.9 to 2.13 are confusing. I understand that LAN001 shows closer affinities with northern populations, as visible in Table 2.9, and therefore only those are tested in Table 2.11. But for MBG004 and MBG016, only Tables 2.12 and 2.13 explore their genetic affinities, directly with southern European populations, but they are not tested with the broad dataset. It seems like a part of the rationale is missing, and should be clarified. Also, what is Table 2.10? Why is it here and to test what with SGermany_EMA...?

- In paragraph 2, the link between CWE and EFF is not clearly explained, it comes much later in Supplementary Note 4. I think a quick explanation here would be useful for the reader, or at least a call to Supplementary Note 4.

- In paragraph 3, WBI is mentioned for the figure 2.8, but no WBI ancestry is shown in this figure...

Isotopes

- MBG010: where do the values of $\delta^{18}O$ used to discuss the southern European coastal origin for this individual come from? There is no reference cited and the oxygen data are still quite scarce nowadays. Could these values be consistent with other regions of Europe?

Height

- Interesting data, but they are not discussed at all, neither here nor in the main text... What about this specific high height for these individuals? Could it be a selection at some point at the beginning of this dynasty, that would justify the power for a few men with specific acts of arms? There is of course no way to reply to this, but it would be great to discuss it a bit more, and maybe to mention in the main text as well.

Supplementary Note 3: Latent pedigree modelling

- I'm not fully convinced by the Bayesian modelling approach here. Not in terms of the model itself, it sounds robust, as far as I can judge it, but I don't think such a model can explain or solve the diversity of situations that are possible in a human society. For instance, the authors don't even consider the possibility of the avuncular relationship on the other around, APG001 being the maternal uncle of HOC001. Of course, it doesn't seem the most parsimonious scenario, but it's not impossible, if we consider wide intervals between APG001 and HOC001's mother.

I'm a bit playing devil's advocate, but I feel this approach is losing a lot of the human parameter, especially as based on a single case. Maybe if it was a repetitive scenario over many relationships, it would look more robust.

This being said, it doesn't remove anything to the proposed interpretations, and in my opinion, the matrilineal system looks strong, however we interpret the kinship between these two individuals, and the overall discussion is really convincing.

Supplementary Note 4: The formation of the Hallstatt Gene pool

General population genetic affinities

- Figure 4.2: The caption is confusing: what is a)? there is no a) or b) in the figure itself, we don't know what the caption refers to.
- Why different outgroups are used for the F3 and the F4 statistics (YRI and CHB)?
- What is the colour legend in the tables 3.1 to 3.5?
- About the figure 4.3, the authors mention the group SGermany_EIA in the text, but the group itself is missing from the table 3.6. How did they plot it on the PCA?

Supervised ADMIXTURE modelling

- The last sentence of the paragraph is unclear: the CWE component is not major in all the regions where Celtic-speaking groups were settled, as shown in Figure 4. What do the authors mean here?

Supplementary Note 5: Population genetic changes after the Hallstatt period

- Which outgroup has been used in the F4 statistics? In the text and in the Figure 5.1, it's written CHB, in the caption of Figure 5.1, it's YRI...
Why is Ireland chosen in the F4 statistics? Is it because that's where the highest frequencies of steppe ancestry are found? It would be good to write it.
- How is built the PCA on Figure 5.2? And what are the arrows in b, c and d?

Reviewer #2:

Remarks to the Author:

The paper presents novel aDNA results from a well-defined group of archaeological sites (primarily a series of exceptionally richly-furnished funerary monuments) dating to the Hallstatt D period in Baden-Württemberg. It convincingly demonstrates the importance of biological relatedness in creating the elite communities represented in these funerary monuments. Of particular importance is the evidence for biological relatedness between geographically distant funerary monuments. This significant result sheds light on social organisation in the Early Iron Age of the region.

Although the overall number of new aDNA samples is small (31 individuals) and one might have wished for a more extensive study, it nonetheless encompasses several of the key sites of the West Hallstatt world, and it is not unreasonable to use these as a basis for tentative generalisation regarding the broader nature of Hallstatt social organisation.

Overall, the methodology appears sound and well documented both in the text and in the very useful figures and extensive supplementary materials.

There are some points of detail that I would like to see addressed before publication:

1. The reference to 'the emergence of Celtic civilisation' in the abstract is somewhat jarring and at odds with the more nuanced discussion of the cultural background in the main text of the paper. I

suggest it be rephrased. In any case, the paper itself makes clear that the hierarchical systems of Hallstatt D collapse around 450 BCE, and cannot thus represent the 'emergence' of anything.

2. The term 'kin groups' in the abstract might more accurately be termed 'biologically-related groups'.

3. The first two sentences of the Introduction somewhat conflate the Hallstatt and La Tène periods (the bulk of the linguistic and historical data relates only to the latter). These two periods are characterised by a very different archaeological signature (for example, the super-rich burials and fortified centres of Hallstatt D essentially disappear in the latter period) and the La Tène period is not addressed directly in the paper, since all of the analysed samples date to Hallstatt D. It would thus be better to focus the text entirely on Hallstatt D: it is still fair to point out that this period has been widely identified with the historical Celts, but it needs to be a little clearer that the use of this ethnonym is problematic (retaining the reference to the Collis 2003 book which deals with the issue in great depth).

4. Recently published aDNA analysis of Hallstatt D individuals from the tumuli at Dolge njive, Slovenia has already shown close familial connections between individuals buried in the same barrow (notably Barrow 1, with a father and four of his children as well as two other close biological relatives). It is surprising that this paper is not referenced or discussed - Armit, I., Fischer, C., Koon, H., Nicholls, R., Olalde, I., Rohland, N., Buckberry, J., Montgomery, J., Mason, P., Črešnar, M., Büster, L., and Reich D. 2023. Kinship practices in Early Iron Age southeast Europe: genetic and isotopic analysis of burials from the Dolge njive barrow cemetery, Dolenjska, Slovenia. *Antiquity*. 97(392): 403-18. <https://doi.org/10.15184/aqy.2023.2>

5. On page 5, paragraph 3, it would be useful to make more emphasis of the observation that the new aDNA findings confirm biological relationships between secondary and central burials. It is not self-evident from the archaeological evidence alone that this would necessarily be the case: in the absence of aDNA evidence, a number of other relationships between central and secondary burials might have been suggested (servants, clients etc).

6. The suggestion of 'totenfolge', also on page 5, is unsupported by any evidence and should best be removed. There are a great many secondary burials at the Magdalenenberg, ranging from very rich to unfurnished, and no convincing reason is presented for singling this one (MBG005) out.

7. The possible pedigree construction for the Asperg and Grafenbuhl individuals appears to assume from the outset that they share a very recent female ancestor. This is based on the shared mt haplotype. However, if matrilineal principles are at work more widely in this society (as seems very plausible) then the elite group may be dominated by a small number of mt haplotypes – in such a case, it seems entirely possible that the two analysed individuals may be, for example, grandfather and grandson, where both of their mothers share the same mt haplotype without themselves necessarily being close biological relatives. This possibility might usefully be considered, and may affect the confidence attributed to the uncle-nephew relationship proposed in the paper.

8. The first sentence of the Discussion cites a 1960 source on social stratification. This seems rather out of date, particularly given the recent prominence of alternative accounts of past social complexification presented in works such as Graeber and Wengrow's 'Dawn of Everything' (2021). The latter, or equivalent should be cited for balance and the opening sentence of the Discussion nuanced

somewhat.

Overall, this is an important and exciting piece of work and, once these points are addressed, I recommend publication of the paper.

Author Rebuttal to Initial comments

Reviewer #1:

The manuscript NATHUMBEHAV-23082609 by Gretzinger and collaborators entitled “Evidence for dynastic succession among early Celtic elites” provides new ancient wide-genome and isotopic data for 31 individuals from Iron Age monumental sites encompassing princely burials in the modern-day southern Germany. These famous sites have been extensively studied in an archaeological point of view and this study provides new multi-proxy results to interpret the interconnectivity of the sites, and therefore of the groups, through genetic relationship as far as 100 km apart, providing insight into matrilineal dynastic systems. This is a great study showing how the development of ancient DNA methods can provide new insights for the understanding of famous sites, really well known archaeologically.

Despite some minor comments and questions that I list below, I enthusiastically support this paper for publication, and I believe Nature Human Behaviour journal is a great support for such studies and I wish more will follow in the future.

Main point:

My main remark about the submitted paper is the lack of integration of the multi-proxy data. Ancient DNA data are really well used, whereas isotopic data, although also produced as part of this work, are forgotten at the beginning (no mention in the abstract and in the first part of the results). On another hand, archaeological data are also a bit missing in the interpretation and discussion: if the main elements are used (like southern connections in the artefacts found in the site), there is no proper discussion linking genomics, isotopes, and archaeology at the individual level. For instance, are there any artefact linking the related individuals? Are the individuals with a specific ancestry buried with specific artefacts that would link them to their area of origin (for instance LAN001)? Do the individuals sharing different ancestries also share similar artefacts? If so, is consistent with a specific geographic location? If not, does it mean the general network is culturally too homogenised? I think the paper would gain from a more integrated discussion with all the elements available, which are numerous and of great quality.

Response: *We have added additional analysis to investigate potential links between Isotopes values, genetics and grave goods. This analysis was added to the Supplementary text*

(“Comparing archeological isotopic and aDNA data” in Supplementary Section 2) and also discussed in the main text (omitting references here): “Furthermore, we do not find a significant association between grave goods, $\delta^{18}\text{O}$ and ancient DNA as markers for nonlocal origin (Supplementary Note 2). For that, we focused on the Magdalenenberg site where a large number of graves exhibit artefacts of transalpine, south-European (especially North Italian and/or southeast Alpine) provenance, indicating cultural transfer alongside extensive, continuous individual-based mobility. We grouped individuals, for which both isotopes and aDNA data were available, into two groups based on the presence of southern, nonlocal artefacts. We find that nonlocal artefacts (being present in 6 out of 16 graves) are not statistically significantly correlated with either higher proportions of EEF ancestry (Two-sided Wilcoxon rank sum exact test; $W = 23$, $p = 0.4923$) nor $\delta^{18}\text{O}$ values (Two-sided Wilcoxon rank sum exact test; $W = 44.5$, $p = 0.1283$) (Supp. Fig. 2.11), both indicating cisalpine origin. Consequently, southern grave goods do not constitute a reliable marker of south-European origin in the Magdalenenberg population, although we do identify individuals with such origins in the burial mound via our isotopic and aDNA data. This is especially evident in the case of MBG010, an adult female, who exhibits $\delta^{18}\text{O}$ and $^{87}\text{Sr}/^{86}\text{Sr}$ values indicative of a northern Italian or Iberian origin, yet is neither buried with southern grave goods nor shows excess genetic affinity to those regions (Supp. Fig. 2.8 & 2.9).”

Minor points:

Main text Introduction

- As mentioned before, it should appear in the abstract that new data are also available for isotopic analysis.

R: *Indeed. We have added this now to the abstract: “Here, we present genomic and isotope data from 31 individuals from this context.”*

- In the introduction, I think some details about the actual chronological relation between both periods Hallstatt and La Tène are missing, while both cultures are discussed and used later in the population genetic section.

R: *We have added absolute dates right into the beginning of the introduction now.*

- The expression “historical sources” at the beginning of the introduction is confusing: do the authors mean actual sources from history (like Caesar maybe?) but then the reference is

missing, or it's a source from an historian, as the cited references suggest it, but then it should be rephrased.

R: *We meant actual primary sources. We clarified now (omitting references here) “While there is scarce historical mention of this ethnonym for the area northwest of the Alps already in the late 6th century BC, there is abundant written evidence from the Greek and Roman spheres that identify the societies associated with the La Tène culture as ‘Celtic’ or ‘Gallic’”*

- The sentence starting with “While there are other forms of kinship, ...” is, in my opinion, incorrect, or maybe not well phrased. Dynastic system can also work through adopted children, for instance in the neighbouring Roman Empire few centuries later. I agree that we will probably never be able to demonstrate it with our tools, especially genomics which obviously will always show biological kinship, but the sentence reads like if dynastic systems can only work with blood lines, which is not true. I would rephrase this.

R: *We agree. We reformulated this now: “A central aspect of a dynastic system of hereditary power is biological relatedness. While there are other forms of kinship, including social relatedness such as fosterage or adoption, which are notoriously difficult to infer from burial archaeology, biological relatedness can be conclusively reconstructed using genetic data.”*

Results

- There is no mention of the new isotopic results generated for this study, as stated further. They need to appear somewhere, especially as they are used later in the result section.

R: *We have added this information to the abstract (see above) as well as to the results section (end of first paragraph): “In addition to genome-wide sequences, we measured $\delta^{18}\text{O}$ and $87\text{Sr}/86\text{Sr}$ values for the 17 of those individuals for whom so far no isotope data had been available, to reconstruct patterns of individual mobility (Supplementary Note 2).” Furthermore, we have added sample size information (from Supp. Table 1.2) to Supplementary Note 2.*

- To help the reader to follow the different explanations about the relationships between the individuals, I suggest to systematically provide their sex and age when there are mentioned for the first time. For instance, it's not specified for HOC003, while it is for the female MBG009. It would help to follow the text better.

R: *We have added the genetic sex to all relevant (explicitly mentioned) individuals in the text,*

either immediately following the label, or in the sentence following. Additionally, all (genetic/osteological) sex and (osteological) age information is reported in Supp. Table 1.1.

- At the end of the fourth paragraph, the mention of the grave of MBG005 and the hypothesis provided is quite interesting, and would benefit of a more detailed explanation: would that case be a sacrifice? of a slave? Or could it be a voluntary death (like the sati in India)? What do the texts say about this? And how common was it? Is this grave the only example found in the sites under study?

R: We share the referee's curiosity, but also highlight that such scenarios can be discussed in a speculative manner at best. In fact, referee 2 voted for removing such speculation entirely, and we now followed this advice. We have deleted this segment.

- In the fifth paragraph, I would refer to the supplementary more precisely, as most of it becomes clear and understandable once we've read the Supplementary Notes 2 and 4. Also, I don't understand why the isotopic results are not discussed here, as some individuals show signal from Italy and Iberia...

R: We highlight now Supplementary Note 2 when the Isotope results are first mentioned and at relevant positions within the text. We further now highlight the presence of MBG010 explicitly in the first Result section, where we write towards the end of the first Results section (slightly shortened here): "Furthermore, we do not find a significant association between grave goods, $\delta^{18}\text{O}$ and ancient DNA as markers for nonlocal origin (Supplementary Note 2). [...] We find that nonlocal artefacts [...] are not statistically significantly correlated with either higher proportions of EEF ancestry [...] nor $\delta^{18}\text{O}$ values [...], both indicating cisalpine origin. Consequently, southern grave goods do not constitute a reliable marker of south-European origin in the Magdalenenberg population, although we do identify individuals with such origins in the burial mound via our isotopic and aDNA data.

This is especially evident in the case of MBG010, an adult female, who exhibits $\delta^{18}\text{O}$ and $^{87}\text{Sr}/^{86}\text{Sr}$ values indicative of a northern Italian or Iberian origin, yet is neither buried with southern grave goods nor shows excess genetic affinity to those regions (Supp. Fig. 2.8 & 2.9)."

- In the last paragraph of the results, out of the 31 individuals, and given the overall ROH values for the group, I would not call "common" the two cases of consanguinity detected here.

R: We agree and have toned this down. We have changed the sentence to "Given that such

high levels are very rare in the published record, the presence of two consanguineous individuals in the comparably small sample size of 30 individuals may suggest that consanguinity was more frequent among the Hallstatt elites of southwestern Germany than in other ancient societies in the archaeogenetic record.”. Note that we also write in the Discussion “In the ancient DNA record, first cousin mating is exceedingly rare, with less than 3% of ancient individuals showing runs of homozygosity consistent (but not conclusive) for the offspring of first cousins (ref)”, which provides a context for our finding of two first-cousins in a sample of 30.

Emergence and decline of the West-Hallstatt gene pool

- The third paragraph shows a nice example of integrated data giving a consistent lifetime story, that's very interesting. I would suggest splitting the second sentence a bit, to make it easier to digest...

R: *As suggested, we have splitted the sentence to make it more reader-friendly.*

- Figure 4. Several problematic points here. The dates are given in calBP, when all the text is written in BCE. Please make it consistent, preferably in BCE, as it's the way it is used in historical times. Panel 4c is hard to understand, many elements are unclear or not explained at all: What are the pie charts representing? Is it all Germany? They need some legend or title. The sources need to be detailed somewhere. The figure needs to be understandable with going to the Supplementary, where the definition of the acronyms is actually given. If some make sense (like NOR), some really do not. What are the symbols in blue and yellow? They are confusing with the symbols given to the sites. This must be explained in the caption.

R: *We have changed all dates in figure 4 from BP to BC/CE. We have modified the figure to make panel C more comprehensible (e.g. we have added a map to visualize the geographic origin of samples used for qpWave and ADMIXTURE analysis). We have also modified and extended the figure caption, where we now also describe the ADMIXTURE clusters used.*

Discussion

- At the end, we would expect a deeper discussion: the paper is about kinship and population structures within a given territory. Do we see any changes in the archaeological data and sites consistent with this increase of Steppe ancestry that seems to be linked with the arrival of the Germanic-speaking tribes? What about these princely burials? These regional structures? And what about the articulation between the Iron Age cultures Hallstatt and La

Tène and the biological signals? More discussion is needed here.

R: *More discussion is always possible, but we believe the paper's scope is already at the limits of this format. As with many studies that present new data for the first time, much space is needed for describing the results and its immediate implications. With respect to the increase of Steppe ancestry in the early middle ages, we do believe that this is an interesting result, but without novel primary data from this period, we can at best be speculative about its immediate relation to specific historical events. We think it suffices to describe the increase on the genetic side, and only suggest possible connections to history. In light of the questions towards the end of the referee's comment, we have added to the discussion. In particular, the second half of the first paragraph is mostly new, where we discuss matrilineality in the broader context of the literature. Also, the second half of the second paragraph is new, now discussing limits of our inference from Southwestern Germany to the broader Hallstatt sphere.*

Methods

- In the contamination estimation section, the second sentence is incomplete. Did you use Schmutzi only? Or a second method to estimate contamination on the mitochondrial DNA?

R: *As in most aDNA papers, we measure mtDNA contamination for all individuals with Schmutzi and for male individuals also on the nuclear genome using ANGSD; this is established convention. ContaMix was also applied, the results are however highly similar to Schmutzi and thus not reported here. Furthermore, we have inspected the damage patterns and fragment lengths of all libraries to verify the authenticity of the retrieved sequences. For samples that showed signals of contamination, we used PMD tools to filter out sequences without aDNA damage. Afterwards, we inspected the individuals again with ANGSD. This is reported in the Methods section.*

- Where has the isotope analysis been performed?

R: *Isotope analysis was conducted at the Curt-Engelhorn-Center Archaeometry gGmbH, Mannheim, Germany. We have added this information to the Methods section.*

Supplementary Information

Some typos have been detected all along the supplementary document, I suggest to carefully double-check it (for instance misspelling of La Tène, found as Latène or La Téne, or the article a/an improperly used). Also it will be necessary to double-check the layout.

R: *We corrected the misspelling and hope to have removed all other typos.*

Supplementary Note 1: Site descriptions

The figures are not properly named in the text and are not called when necessary (for instance p12). All this needs to be double-checked.

R: *Indeed. This has now been fixed. All figures are now explicitly referenced within the text.*

Supplementary Note 2: Kinship, Inbreeding and individual ancestry Kinship

- typo at the third line: "als"
- typo in paragraph 3: "Again."

R: *Thanks, we have corrected this typo.*

- Do the authors have any explanation why the coefficient of relatedness for APG001 and HOC001 is higher than expected for a second-degree relationship? Could it be linked to a background relatedness?

R: *For APG001, we do not identify RoH indicative of recent parental relatedness that might result in excess related to HOC001. However, HOC001 is of relatively low-coverage which adds noise to the pairwise mismatch rates (PMMR) that underlie this analysis. This is evident in the normal distribution of each degree of relatedness given the number of overlapping SNPs, as shown in Supp. Fig. 2.3. Still, the PMMR value falls within the range expected for second-degree relatives with the available SNP overlap.*

- Figure 2.1: The symbol and colour for HOC003 are wrong.

R: *We have corrected the figure and changed the symbol.*

- Figure 2.3: In the legend, it should be specified that these plots are built following the binomial distribution test from BREADR.

R: *We have added this information to the figure caption.*

Inbreeding

- It might be useful to repeat here the threshold of 300,000 SNPs used in hapROH.

R: *We have added this to the text.*

- Figure 2.7: APG003 and MBG004 are both interpreted as first-cousin's offspring, while the plots b and d show very different patterns. The plot b shows a scenario where the inbreeding seems to be consistent with one between closer parents, like full siblings. Could the authors comment on these results and their interpretations?

R: *The detected RoH in APG003 and MBG004 are, based on their length and number, only consistent with first-cousin inbreeding. Closer scenarios of inbreeding would result in substantially longer RoH (>400cM). The plots with the model fits (Supp Fig. 2.7b,d) can be misleading due to the discrete nature of the RoH counts and classes and the continuous distribution shown.*

Individual ancestry

- Tables 2.9 to 2.13 are confusing. I understand that LAN001 shows closer affinities with northern populations, as visible in Table 2.9, and therefore only those are tested in Table 2.11. But for MBG004 and MBG016, only Tables 2.12 and 2.13 explore their genetic affinities, directly with southern European populations, but they are not tested with the broad dataset. It seems like a part of the rationale is missing, and should be clarified. Also, what is Table 2.10? Why is it here and to test what with SGermany_EMA...?

R: *Both the northern affinity of LAN001 and the southern affinities of MBG004 and MBG016 are visible from PCA, ADMIXTURE and qpAdm analyses (as described various earlier places in the main text and Supplement). For all three individual outliers, tables 2.9 through 2.13 then describe targeted affinity tests based on F4 and - where possible given the limited genetic variation among the sources - also qpWave statistics, specifically restricting to plausible source populations from the North and South, respectively.*

Table 2.10 shows the affinity of the medieval population of southern Germany, highlighting that this affinity is different to the signal observed in LAN001, evidencing closer genetic similarity to Northern German and Scandinavian groups. This is mentioned in the main text in section "Emergence and decline of the West-Hallstatt gene pool" and the table was placed there due to the shared methodological approach.

- In paragraph 2, the link between CWE and EFF is not clearly explained, it comes much later in Supplementary Note 4. I think a quick explanation here would be useful for the reader, or at least a call to Supplementary Note 4.

R: We have added an explanatory segment to the section explaining the relationship between CWE and EEF. “(..) This is expected since CWE ancestry is enriched with EEF ancestry due to the higher proportion of Early European farmer ancestry in southern and western Europe compared to Europe north of the Alps (where EEF-depleted, Steppe-ancestry enriched CNE and NOR ancestry components prevail).”

- In paragraph 3, WBI is mentioned for the figure 2.8, but no WBI ancestry is shown in this figure...

R: While we do not plot the individual WBI ancestry, all estimates for published and novel samples can be found in Supp. Table. 3.7. We have moved the reference to the mentioned figure further below, and instead now refer to Supp. Table 3.7.

Isotopes

- MBG010: where do the values of $\delta^{18}\text{O}$ used to discuss the southern European coastal origin for this individual come from? There is no reference cited and the oxygen data are still quite scarce nowadays. Could these values be consistent with other regions of Europe?

R: We have added the citations from which the references for the areas were taken, they were originally established in Oelze et al. 2012. The combination of $\delta^{18}\text{O}$ and $^{87}\text{Sr}/^{86}\text{Sr}$ are consistent with various coastal areas in Iberia, Italy, and France, yet, not with southern Germany and neighboring regions.

Height Interesting data, but they are not discussed at all, neither here nor in the main text... What about this specific high height for these individuals? Could it be a selection at some point at the beginning of this dynasty, that would justify the power for a few men with specific acts of arms? There is of course no way to reply to this, but it would be great to discuss it a bit more, and maybe to mention in the main text as well.

R: Indeed. We have now added a segment about the height data to the main text: “The close biological relationship between the two may also explain their exceptional body heights. While male individuals from elite graves are already significantly taller than males from secondary burials (Two-sided Wilcoxon rank sum exact test; $W = 67$, $p = 0.004067$), HOC001, followed by his relative APG001, are the tallest individuals in the complete osteological record of Iron Age southern Germany (Supp. Fig. 2.10). This highlights the possibility that, besides better nutrition, also genetic relatedness may have contributed to this social differentiation in body height.”

We note that previously published research has shown that some Bronze and Iron Age elite burials, e.g. the prince of Glauberg, had a more meat-rich subsistence than the non-elite population, which might have favored their growth (cf. Knipper et al. 2014). Yet, genetic relatedness between the individuals might be another factor, spreading the genetic predisposition for tall body height in the elite population.

Supplementary Note 3: Latent pedigree modelling

- I'm not fully convinced by the Bayesian modelling approach here. Not in terms of the model itself, it sounds robust, as far as I can judge it, but I don't think such a model can explain or solve the diversity of situations that are possible in a human society. For instance, the authors don't even consider the possibility of the avuncular relationship on the other around, APG001 being the maternal uncle of HOC001. Of course, it doesn't seem the most parsimonious scenario, but it's not impossible, if we consider wide intervals between APG001 and HOC001's mother.

I'm a bit playing devil's advocate, but I feel this approach is losing a lot of the human parameter, especially as based on a single case. Maybe if it was a repetitive scenario over many relationships, it would look more robust.

This being said, it doesn't remove anything to the proposed interpretations, and in my opinion, the matrilineal system looks strong, however we interpret the kinship between these two individuals, and the overall discussion is really convincing.

R: *First, we have substantially extended this modelling approach now to a more complete set of models, including parental and grandparental models as well as reverse cases as suggested, with APG001 being HOC001's uncle etc. In total, we now test 11 models and extend the approach to now also include likelihoods for maternal background relatedness, to be able to incorporate models without implicit mitochondrial match. It turns out, that our new analysis still ranks the maternal-avuncular model (HOC001's sister being APG001's mother) the highest (86%), but now includes a maternal grandfather model as a second-best hit (6%), generally consistent with a matrilineal system of inheritance for this particular group.*

We have rewritten and much extended Supplementary Section 3 and the related summary in the main text as well as Figure 2. We also now write in the discussion (on page 13): "Yet, we highlight that this leadership system may be limited to southern Germany and not apply to the rest of the Hallstatt sphere. In addition, there might be differences between the elite and the larger common population."

Supplementary Note 4: The formation of the Hallstatt Gene pool General

population genetic affinities

- Figure 4.2: The caption is confusing: what is a)? there is no a) or b) in the figure itself, we don't know what the caption refers to.

R: *We have modified and extended the figure caption to make it easier to follow. It reads now: "Supplementary Figure 4.2. Principal Component Analysis of European variation and ancient population structure. Genetic structure of published and novel ancient individuals in this study, projected onto figure 4.1. The left plot shows several Iron Age individuals plotted together with our novel samples from early Iron Age southern Germany. In the right plot, the ancient reference groups are only shown as convex hulls for better visibility. Novel Iron Age and previously published Middle Bronze Age samples from southern Germany are represented with individual symbols. The symbols and colours correspond to Figure 1. Outlier individuals mentioned in the main text are labelled."*

- Why different outgroups are used for the F3 and the F4 statistics (YRI and CHB)?

R: *We have used African outgroups (Mbuti.DG and YRI.SG) where possible. In analyses that directly involved African populations or populations with detectable African admixture, we used Han Chinese (CHB.SG) as outgroup to avoid outgroup attraction.*

- What is the colour legend in the tables 3.1 to 3.5?

R: *The colour code for these tables is described in each caption of the respective tables. Central and western European Test populations are marked in green, southern European populations in orange, northern European populations in darkblue, British and Irish populations in lightblue, and northeastern European populations in purple.*

- About the figure 4.3, the authors mention the group SGermany_EIA in the text, but the group itself is missing from the table 3.6. How did they plot it on the PCA?

R: *We cannot really reproduce this, at least SGermany_EIA (all of our novel genomes pooled together) is part of table 3.6 and plotted in Supp. Fig. 4.3 and labeled with bold letters. It plots closest to Slovenia_LBA, Hungary_LBA, France_GrandEst_IA1 and Hungary_EIA_Hallstatt, falling in between the diversity of Bronze Age/Iron Age France and Bronze Age/Iron Age Hungary and Slovenia.*

Supervised ADMIXTURE modelling

- The last sentence of the paragraph is unclear: the CWE component is not major in all the regions where Celtic-speaking groups were settled, as shown in Figure 4. What do the authors mean here?

R: *We reformulated this statement to be more precise. We now write: “In general, we do not find one homogeneous gene pool that covers the whole geographical area that is associated with the Celtic languages and the spread of the Hallstatt and La Tène cultures. While the CWE component is the largest ancestry component in most populations that inhabited these regions from the Middle Bronze Age to the end of the Iron Age and Roman period, indicating a shared biological background for all those groups, we however observe strong genetic variation that is structured geographically (Figure 3).”*

Supplementary Note 5: Population genetic changes after the Hallstatt period

- Which outgroup has been used in the F4 statistics? In the text and in the Figure 5.1, it's written CHB, in the caption of Figure 5.1, it's YRI...

Why is Ireland chosen in the F4 statistics? Is it because that's where the highest frequencies of steppe ancestry are found? It would be good to write it.

R: *We have corrected the figure caption, the used outgroup was CHB. Indeed, Ireland was used because of its high percentage of Steppe ancestry and phylogenetic position. We have added information on this to the supplementary text. It reads: “Ireland was used for comparison due to its high proportion of Steppe and low amount of WHG ancestry.*

Additionally, we assume that most ancient samples from Germany should be, depending on their ancestry, closer related to either continental southern European or continental Northern European, highlighting changes in the local ancestry due to influx from the South or the North.”

- How is built the PCA on Figure 5.2? And what are the arrows in b, c and d?

R: *We have added information how the PCA on the EEF, WHG, and Steppe ancestry populations estimates was constructed to the Methods section and the supplementary text. In the methods we now write: “PCA on the Steppe, WHG, and EEF components measured in 153 ancient and present-day populations was calculated using the prcomp function from the stats package (v3.6.2) in R (v4.1.1).”. We removed the arrows, which indicated PC1 and PC2, to enhance the readability of the figure.*

Reviewer #2:

Remarks to the Author:

The paper presents novel aDNA results from a well-defined group of archaeological sites (primarily a series of exceptionally richly-furnished funerary monuments) dating to the Hallstatt D period in Baden-Württemberg. It convincingly demonstrates the importance of biological relatedness in creating the elite communities represented in these funerary monuments. Of particular importance is the evidence for biological relatedness between geographically distant funerary monuments. This significant result sheds light on social organisation in the Early Iron Age of the region.

Although the overall number of new aDNA samples is small (31 individuals) and one might have wished for a more extensive study, it nonetheless encompasses several of the key sites of the West Hallstatt world, and it is not unreasonable to use these as a basis for tentative generalisation regarding the broader nature of Hallstatt social organisation.

Overall, the methodology appears sound and well documented both in the text and in the very useful figures and extensive supplementary materials.

There are some points of detail that I would like to see addressed before publication:

1. The reference to ‘the emergence of Celtic civilisation’ in the abstract is somewhat jarring and at odds with the more nuanced discussion of the cultural background in the main text of the paper. I suggest it be rephrased. In any case, the paper itself makes clear that the hierarchical systems of Hallstatt D collapse around 450 BCE, and cannot thus represent the ‘emergence’ of anything.

R: *We changed the beginning of the abstract (which is now also shortened according for format guidelines) substantially. It now reads: “The early Iron Age in France, Germany, and Switzerland, known as the West-Hallstattkreis, stands out as featuring the earliest evidence for supra-regional organisation north of the Alps. Often referred to as ‘early Celtic’, suggesting tentative connections to later cultural phenomena, its societal and population structure remain enigmatic. Here, we present”*

2. The term ‘kin groups’ in the abstract might more accurately be termed ‘biologically-related groups’.

R: *We agree and have changed it as suggested.*

3. The first two sentences of the Introduction somewhat conflate the Hallstatt and La Tène periods (the bulk of the linguistic and historical data relates only to the latter). These two periods are characterised by a very different archaeological signature (for example, the super-rich burials and fortified centres of Hallstatt D essentially disappear in the latter period) and the La Tène period is not addressed directly in the paper, since all of the analysed samples date to Hallstatt D. It would thus be better to focus the text entirely on Hallstatt D: it is still fair to point out that this period has been widely identified with the historical Celts, but it needs to be a little clearer that the use of this ethnonym is problematic (retaining the reference to the Collis 2003 book which deals with the issue in great depth).

R: We agree that the concept of Celtic as ethnonym is problematic, as is the identification of archaeological "cultures" with such ethnonyms. We have now rewritten the beginning of the introduction to be more nuanced. With respect to the premise of a discontinuity between Ha D and Lt A, we tend to disagree. Specifically, princely tombs in south-west Germany, for example at the Ipf, the Hohenasperg, the Breisacher Münsterberg, the Glauberg and other early Celtic centres of power, continue to exist up to Lt A, covering the period between approx. 600 and 400 BC, thus bridging Ha D and Lt A. Nevertheless, our new text now makes it clear that there is certainly a difference in the degree of description as "Celtic" between Hallstatt and La Tène. The first paragraph of the introduction now reads: "The European Iron Age north of the Alps is characterised by the two key archaeological cultures Hallstatt (800 - 450 BCE) and La Tène (after 450 BCE until the beginning of the Roman period around 50 BCE), which have been, to a different degree, described as 'Celtic' (refs).

While there is scarce historical mention of this ethnonym for the area northwest of the Alps already in the late 6th century BC, there is abundant written evidence from the Greek and Roman spheres that identify the societies associated with the La Tène culture as 'Celtic' or 'Gallic' (refs). Albeit being problematic as an ethnonym, these descriptions highlight the close connection between a specific archaeological horizon, hypothesised linguistic affiliations (the Celtic languages), and historical sources (refs). The pan-European patterns and linguistic evidence for cultural connections during this time are complex and encompass a vast region from the Iberian Peninsula and the British Isles throughout Central Europe and as far east as Anatolia (during the 3rd century BCE). While older research assumed an exclusive emergence of this later pan-European phenomenon in a relatively narrowly defined area northwest of the Alps, newer perspectives suggest a model of polycentric emergence in a wide area between the Atlantic coast and southwestern Germany (ref). One of these core regions was located in present-day eastern France, Switzerland and southwestern Germany. Between 600 to 400 BCE (Hallstatt D and La Tène A) this area stands out in its archaeological importance, as highlighted by rich 'princely' burials ('Fürstengräber')."

4. Recently published aDNA analysis of Hallstatt D individuals from the tumuli at Dolge njive, Slovenia has already shown close familial connections between individuals buried in the same barrow (notably Barrow 1, with a father and four of his children as well as two other close biological relatives). It is surprising that this paper is not referenced or discussed - Armit, I., Fischer, C., Koon, H., Nicholls, R., Olalde, I., Rohland, N., Buckberry, J., Montgomery, J., Mason, P., Črešnar, M., Büster, L., and Reich D. 2023. Kinship practices in Early Iron Age southeast Europe: genetic and isotopic analysis of burials from the Dolge njive barrow cemetery, Dolenjska, Slovenia. *Antiquity*. 97(392): 403-18.

<https://doi.org/10.15184/aqy.2023.2>

R: *Indeed, we missed to cite this. We have now added it and discuss this citation in the discussion section: “Recent genetic evidence from the Hallstatt Dolge njive barrow cemetery in Slovenia is neither consistent with a strictly matrilineal nor patrilineal kinship structure for the buried population (ref) and might indicate a more complex heritability system along both the male and female lines that potentially included adoption or fosterage as well (ref).”*

5. On page 5, paragraph 3, it would be useful to make more emphasis of the observation that the new aDNA findings confirm biological relationships between secondary and central burials. It is not self-evident from the archaeological evidence alone that this would necessarily be the case: in the absence of aDNA evidence, a number of other relationships between central and secondary burials might have been suggested (servants, clients etc).

R: *We note that at least for the Magdalenenberg, there is an explicit hypothesis of a so-called “kin group” burial mound. We now added to the mentioned paragraph this sentence with reference: “We note that the biological relatedness detected between the central and secondary burials is consistent with descriptions of the Magdalenenberg as a “kin group” burial mound for an “enlarged family” (Metzner-Nebelsick 2018).”*

6. The suggestion of ‘totenfolge’, also on page 5, is unsupported by any evidence and should best be removed. There are a great many secondary burials at the Magdalenenberg,

ranging from very rich to unfurnished, and no convincing reason is presented for singling this one (MBG005) out.

R: *We agree this was rather speculative and have now removed it.*

7. The possible pedigree construction for the Asperg and Grafenbuhl individuals appears to

assume from the outset that they share a very recent female ancestor. This is based on the shared mt haplotype. However, if matrilineal principles are at work more widely in this society (as seems very plausible) then the elite group may be dominated by a small number of mt haplotypes – in such a case, it seems entirely possible that the two analysed individuals may be, for example, grandfather and grandson, where both of their mothers share the same mt haplotype without themselves necessarily being close biological relatives. This possibility might usefully be considered, and may affect the confidence attributed to the uncle-nephew relationship proposed in the paper.

R: Indeed, good point. This comment led us to extend our modelling approach substantially to a more complete set of models, including parental and grandparental models. In total, we now test 11 models and extend the approach to now also include likelihoods for maternal background relatedness, to be able to incorporate models without implicit mitochondrial match. It turns out, that our new analysis still ranks the maternal-avuncular model (HOC001's sister being APG001's mother) the highest (86%), but now includes a maternal grandfather model as a second-best hit (6%), generally consistent with a matrilineal system of inheritance for this particular case. Please also see our related comment to referee 1.

8. The first sentence of the Discussion cites a 1960 source on social stratification. This seems rather out of date, particularly given the recent prominence of alternative accounts of past social complexification presented in works such as Graeber and Wengrow's 'Dawn of Everything' (2021). The latter, or equivalent should be cited for balance and the opening sentence of the Discussion nuanced somewhat.

R: Thanks. We have added the citation as recommended and reformulated the sentence: "Hereditary leadership is described as one key aspect of early historically recorded complex societies around the world (refs) but it is hard to prove by archaeological record only."

Decision Letter, first revision:

14th February 2024

Dear Dr. Schiffels,

Thank you for your patience as we've prepared the guidelines for final submission of your Nature Human Behaviour manuscript, "Evidence for dynastic succession among early Celtic elites" (NATHUMBEHAV-23082609A). Please carefully follow the step-by-step instructions provided in the attached file, and add a response in each row of the table to indicate the changes that you have made. Please also address the additional marked-up edits we have proposed within the reporting summary.

Ensuring that each point is addressed will help to ensure that your revised manuscript can be swiftly handed over to our production team.

We would hope to receive your revised paper, with all of the requested files and forms within two-three weeks. Please get in contact with us if you anticipate delays.

Nature Human Behaviour offers a Transparent Peer Review option for new original research manuscripts submitted after December 1st, 2019. As part of this initiative, we encourage our authors to support increased transparency into the peer review process by agreeing to have the reviewer comments, author rebuttal letters, and editorial decision letters published as a Supplementary item. When you submit your final files please clearly state in your cover letter whether or not you would like to participate in this initiative. Please note that failure to state your preference will result in delays in accepting your manuscript for publication.

In recognition of the time and expertise our reviewers provide to Nature Human Behaviour's editorial process, we would like to formally acknowledge their contribution to the external peer review of your manuscript entitled "Evidence for dynastic succession among early Celtic elites". For those reviewers who give their assent, we will be publishing their names alongside the published article.

Cover suggestions

We welcome submissions of artwork for consideration for our cover. For more information, please see our guide for cover artwork.

ORCID

Non-corresponding authors do not have to link their ORCIDs but are encouraged to do so. Please note that it will not be possible to add/modify ORCIDs at proof. Thus, please let your co-authors know that if they wish to have their ORCID added to the paper they must follow the procedure described in the following link prior to acceptance:

Nature Human Behaviour has now transitioned to a unified Rights Collection system which will allow our Author Services team to quickly and easily collect the rights and permissions required to publish your work. Approximately 10 days after your paper is formally accepted, you will receive an email in providing you with a link to complete the grant of rights. If your paper is eligible for Open Access, our Author Services team will also be in touch regarding any additional information that may be required to arrange payment for your article.

Please note that *Nature Human Behaviour* is a Transformative Journal (TJ). Authors may publish their research with us through the traditional subscription access route or make their paper immediately open access through payment of an article-processing charge (APC). Authors will not be required to make a final decision about access to their article until it has been accepted. Find out more about Transformative Journals

[REDACTED]

Best regards,
[REDACTED]

On behalf of

[REDACTED]

Reviewer #1:

Remarks to the Author:

This is the second review to the manuscript NATHUMBEHAV-23082609 by Gretzinger and collaborators entitled "Evidence for dynastic succession among early Celtic elites".

I acknowledge the corrections and efforts brought to the manuscript by the authors, which makes a much clearer and more understandable text, as well as more nuanced. The integration of the

multi-proxy data is also better and gives a fairer place to isotopic, archaeological, and anthropological data.

I also appreciate a lot the new analysis and discussion about the biological relatedness models, giving more space to other scenarios that could have happened, more representative of the past societies.

The paper is definitely suitable for publication now.

There are a very few more typos or very minor revisions though, that I detected in the new version and that I list below.

p. 5, 8 and 9 – Several times in the text: “(Two-sided Wilcoxon rank...)” where the first “T” should be lower case.

p.13 – first paragraph of the discussion: “today” twice in the same sentence.

p.14 – paragraph starting with “The early Celtic...”: typo in “at least”.

Suppl. information

Fig. 4.4 – I hadn’t noticed before, but the looking-down triangles for the Neolithic individuals is confusing with the same for the present-day populations. It would be nice to change them, or the colour maybe.

Tables 3.1-3.5 – I now see the colour code written in the headers in these tables. However, it would be better (and easy to do) to add them as extra lines within the sheet to make it more obvious.

Reviewer #2:

Remarks to the Author:

This is an important and convincing paper that presents novel aDNA results from a crucial period and region in European prehistory. The redraft has taken on board the comments of the reviewers on the original draft.

I have just a few, mostly minor, comments on the redraft:

1. the first paragraph of the main text (lines 40-57) is a little over-complicated and could usefully be shortened. The basic point is that the HaD societies studied here have been widely regarded as Early Celtic, although the use of the term 'Celtic' as an ethnonym is now widely recognised to be problematic due to its different usages by archaeologists, linguists and classical historians. It is no problem to refer to the HaD societies as Celtic so long as this basic disclaimer is included.

2. In that same paragraph I would suggest replacing 'already in the late 6th century' with 'before the fourth century' - this would more properly reflect the step change in the quantity and quality of sources that emerges in the period of the Celtic migrations/raids into Italy.

3. Lines 69-70. It is not correct to say that these burials 'represent the first rudiments of the emergence of dynastic systems of power, because, as the authors rightly point out elsewhere, these dynastic systems do not continue after the period concerned. The archaeological evidence of the 4th

and 3rd centuries does not support the continuation of power structures on this scale.

4. Lines 91 onwards. It would help the reader if the number of analysed samples for each site were given in brackets when listing the sites.

5. Line 167. It may be useful to mention that the time gap of c. 100 years between these two individuals linked as 3rd degree relatives would strongly suggest a 'vertical' relationship - perhaps great grandmother-great grandson.

6. Line 169. The mention of the 'Kapf settlement' seems oddly specific - for an isotopic area of origin it would be better to be somewhat less precise, e.g. 'in the region containing the Kapf settlement'.

Author Rebuttal, first revision:

Reviewer #1 (Remarks to the Author):

This is the second review to the manuscript NATHUMBEHAV-23082609 by Gretzinger and collaborators entitled "Evidence for dynastic succession among early Celtic elites".

I acknowledge the corrections and efforts brought to the manuscript by the authors, which makes a much clearer and more understandable text, as well as more nuanced. The integration of the multi-proxy data is also better and gives a fairer place to isotopic, archaeological, and anthropological data.

I also appreciate a lot the new analysis and discussion about the biological relatedness models, giving more space to other scenarios that could have happened, more representative of the past societies.

The paper is definitely suitable for publication now.

There are a very few more typos or very minor revisions though, that I detected in the new version and that I list below.

p. 5, 8 and 9 – Several times in the text: "(Two-sided Wilcoxon rank...)" where the first "T" should be lower case.

p.13 – first paragraph of the discussion: "today" twice in the same sentence.

p.14 – paragraph starting with “The early Celtic...”: typo in “at least”.

Response: *We thank the referee for spotting these. All fixed now.*

Suppl. information

Fig. 4.4 – I hadn’t noticed before, but the looking-down triangles for the Neolithic individuals is confusing with the same for the present-day populations. It would be nice to change them, or the colour maybe.

Response: *We have indeed followed that suggestion and now plot them in red.*

Tables 3.1-3.5 – I now see the colour code written in the headers in these tables. However, it would better (and easy to do) to add them as extra lines within the sheet to make it more obvious.

Response: *Indeed, we have put these in now as extra lines as suggested.*

Reviewer #2 (Remarks to the Author):

This is an important and convincing paper that presents novel aDNA results from a crucial period and region in European prehistory. The redraft has taken on board the comments of the reviewers on the original draft.

I have just a few, mostly minor, comments on the redraft:

1. the first paragraph of the main text (lines 40-57) is a little over-complicated and could usefully be shortened. The basic point is that the HaD societies studied here have been widely regarded as Early Celtic, although the use of the term 'Celtic' as an ethnonym is now widely recognised to be problematic due to its different usages by archaeologists, linguists and classical historians. It is no problem to refer to the HaD societies as Celtic so long as this basic disclaimer is included.

Response: *We have now edited the beginning further to make it hopefully less complicated and easier to read.*

2. In that same paragraph I would suggest replacing 'already in the late 6th century' with 'before the fourth century' - this would more properly reflect the step change in the quantity and quality of sources that emerges in the period of the Celtic migrations/raids into Italy.

Response: *We would like to keep the reference to the 6th century, as that is indeed the first mention of the term "Keltoi" in Greek sources, but we have made the sentence more specific to say exactly that: "Today regarded problematic as an ethnonym, the name 'Celtic' was first mentioned in Greek sources from the late 6th century BC, and it is abundantly used in antique sources for societies associated with the La Tène culture". We think this accurately reflects the step change the referee is referring to.*

3. Lines 69-70. It is not correct to say that these burials 'represent the first rudiments of the emergence of dynastic systems of power, because, as the authors rightly point out elsewhere, these dynastic systems do not continue after the period concerned. The archaeological evidence of the 4th and 3rd centuries does not support the continuation of power structures on this scale.

Response: *We have changed this sentence and in particular removed the term "emergence" to not make it seem as this was the beginning of a continuing trend. The sentence now reads: "Accordingly, those monumental princely burials would represent the manifestation of dynastic systems of power, in which*

political hegemony was at least partially based on biologically inherited privilege [refs], a hallmark of early complex societies [ref].”

4. Lines 91 onwards. It would help the reader if the number of analysed samples for each site were given in brackets when listing the sites.

Response: *Indeed, we have added those numbers now.*

5. Line 167. It may be useful to mention that the time gap of c. 100 years between these two individuals linked as 3rd degree relatives would strongly suggest a 'vertical' relationship - perhaps great grandmother-great grandson.

Response: *Very good point indeed. We have now added the sentence: “Based on the chronological difference between the graves, an ancestral relationship between both individuals (such as great-grandmother and great-grandson) appears most coherent”.*

6. Line 169. The mention of the 'Kapf settlement' seems oddly specific - for an isotopic area of origin it would be better to be somewhat less precise, e.g. 'in the region containing the Kapf settlement'.

Response: *We have followed this suggestion and now write “...in the region around the Kapf, the settlement associated with Magdalenenberg...”*

Final Decision Letter:

Dear Dr Schiffels,

We are pleased to inform you that your Article "Evidence for dynastic succession among early Celtic elites in Central Europe", has now been accepted for publication in Nature Human Behaviour.

Please note that *Nature Human Behaviour* is a Transformative Journal (TJ). Authors may publish their research with us through the traditional subscription access route or make their paper immediately open access through payment of an article-processing charge (APC). Authors will not be required to make a final decision about access to their article until it has been accepted. Find out more about Transformative Journals

We welcome the submission of potential cover material (including a short caption of around 40 words) related to your manuscript; suggestions should be sent to Nature Human Behaviour as electronic files (the image should be 300 dpi at 210 x 297 mm in either TIFF or JPEG format). Please note that such

pictures should be selected more for their aesthetic appeal than for their scientific content, and that colour images work better than black and white or grayscale images. Please do not try to design a cover with the Nature Human Behaviour logo etc., and please do not submit composites of images related to your work. I am sure you will understand that we cannot make any promise as to whether any of your suggestions might be selected for the cover of the journal.

With best regards,

[REDACTED]